# Wavefunction matching for solving quantum many-body problems

Serdar Elhatisari[1,2], Lukas Bovermann[3], Yuan-Zhuo Ma[4,5], Evgeny Epelbaum[3], Dillon Frame[6,7], Fabian Hildenbrand[6,7], Myungkuk Kim[8], Youngman Kim[8], Hermann Krebs[3], Timo A. Lähde[6,7], Dean Lee[4✉], Ning Li[9], Bing-Nan Lu[10], Ulf-G. Meißner[2,6,7,11], Gautam Rupak[12], Shihang Shen[6,7], Young-Ho Song[13] & Gianluca Stellin[14]

Ab initio calculations have an essential role in our fundamental understanding of quantum many-body systems across many subfields, from strongly correlated fermions[1–3] to quantum chemistry[4–6] and from atomic and molecular systems[7–9] to nuclear physics[10–14]. One of the primary challenges is to perform accurate calculations for systems where the interactions may be complicated and difficult for the chosen computational method to handle. Here we address the problem by introducing an approach called wavefunction matching. Wavefunction matching transforms the interaction between particles so that the wavefunctions up to some finite range match that of an easily computable interaction. This allows for calculations of systems that would otherwise be impossible owing to problems such as Monte Carlo sign cancellations. We apply the method to lattice Monte Carlo simulations[15,16] of light nuclei, medium-mass nuclei, neutron matter and nuclear matter. We use high-fidelity chiral effective field theory interactions[17,18] and find good agreement with empirical data. These results are accompanied by insights on the nuclear interactions that may help to resolve long-standing challenges in accurately reproducing nuclear binding energies, charge radii and nuclear-matter saturation in ab initio calculations[19,20].

Quantum Monte Carlo simulations are a powerful and efficient ab initio method for describing quantum many-body systems using stochastic processes[1,9,15,16,21–23]. If the Monte Carlo amplitudes are positive, then the computational effort grows only as a low power of the number of particles. For many problems of interest, a simple Hamiltonian $H^S$ can be found that is easily computable using Monte Carlo methods and describes the energies and other observable properties of the many-body system in fair agreement with empirical data[24–27]. However, realistic high-fidelity Hamiltonians usually suffer from severe sign problems with positive and negative contributions cancelling each other so that Monte Carlo calculations become impractical. Here we solve the problem using an approach called wavefunction matching. While keeping the observable physics unchanged, wavefunction matching creates a new high-fidelity Hamiltonian $H'$ such that the two-body wavefunctions up to some finite range match that of a simple Hamiltonian $H^S$, which is easily computed. This allows for a rapidly converging expansion in powers of the difference $H' − H^S$. Although wavefunction matching can be used with any computational scheme, we focus here on quantum Monte Carlo simulations where the method presents a practical strategy for evading sign oscillations in high-fidelity calculations.

While $H^S$ and $H'$ act on many-body systems, the wavefunction-matching process is done at the two-body level only. For the sake of clarity, we define $H^S$ and $H'$ as containing only two-body interactions. Later we also consider the inclusion of three-body interactions. However, that analysis is separate from wavefunction matching.

A unitary transformation $U$ is a linear transformation that maps normalized orthogonal states to other normalized orthogonal states. Starting from a high-fidelity Hamiltonian $H$ with only two-body interactions, wavefunction matching defines a new Hamiltonian $H' = U^\dagger HU$, where $U^\dagger$ is the Hermitian conjugate of $U$. The unitary transformation is performed at the two-body level. In each two-body angular momentum channel, the unitary transformation $U$ is active only when the separation distance between two particles is less than some chosen distance $R$. For the calculations presented here, the value $R = 3.72$ fm is used. The dependence on $R$ is extensively discussed in Supplementary Information.

Let us write $\psi_0(r)$, $\psi'_0(r)$ and $\psi^S_0(r)$ for the two-body ground-state wavefunctions of $H$, $H'$ and the simple Hamiltonian $H^S$, respectively. Here $r$ is the distance between the two particles. The transformation $U$ is defined such that $\psi'_0(r)$ is proportional to $\psi^S_0(r)$ for $r < R$. The simple

[1]Faculty of Natural Sciences and Engineering, Gaziantep Islam Science and Technology University, Gaziantep, Turkey. [2]Helmholtz-Institut für Strahlen- und Kernphysik and Bethe Center for Theoretical Physics, Universität Bonn, Bonn, Germany. [3]Institut für Theoretische Physik II, Ruhr-Universität Bochum, Bochum, Germany. [4]Facility for Rare Isotope Beams and Department of Physics and Astronomy, Michigan State University, East Lansing, MI, USA. [5]Guangdong Provincial Key Laboratory of Nuclear Science, Institute of Quantum Matter, South China Normal University, Guangzhou, China. [6]Institut für Kernphysik, Institute for Advanced Simulation, Jülich Center for Hadron Physics, Jülich, Germany. [7]Center for Advanced Simulation and Analytics (CASA), Forschungszentrum Jülich, Jülich, Germany. [8]Center for Exotic Nuclear Studies, Institute for Basic Science, Daejeon, Korea. [9]School of Physics, Sun Yat-Sen University, Guangzhou, China. [10]Graduate School of China Academy of Engineering Physics, Beijing, China. [11]Tbilisi State University, Tbilisi, Georgia. [12]Department of Physics and Astronomy and HPC2 Center for Computational Sciences, Mississippi State University, Mississippi State, MI, USA. [13]Institute for Rare Isotope Science, Institute for Basic Science (IBS), Daejeon, Korea. [14]ESNT, DRF/IRFU/DPhN/LENA, CEA Paris-Saclay and Université Paris-Saclay, Gif-sur-Yvette, France. ✉e-mail: leed@frib.msu.edu

# Article

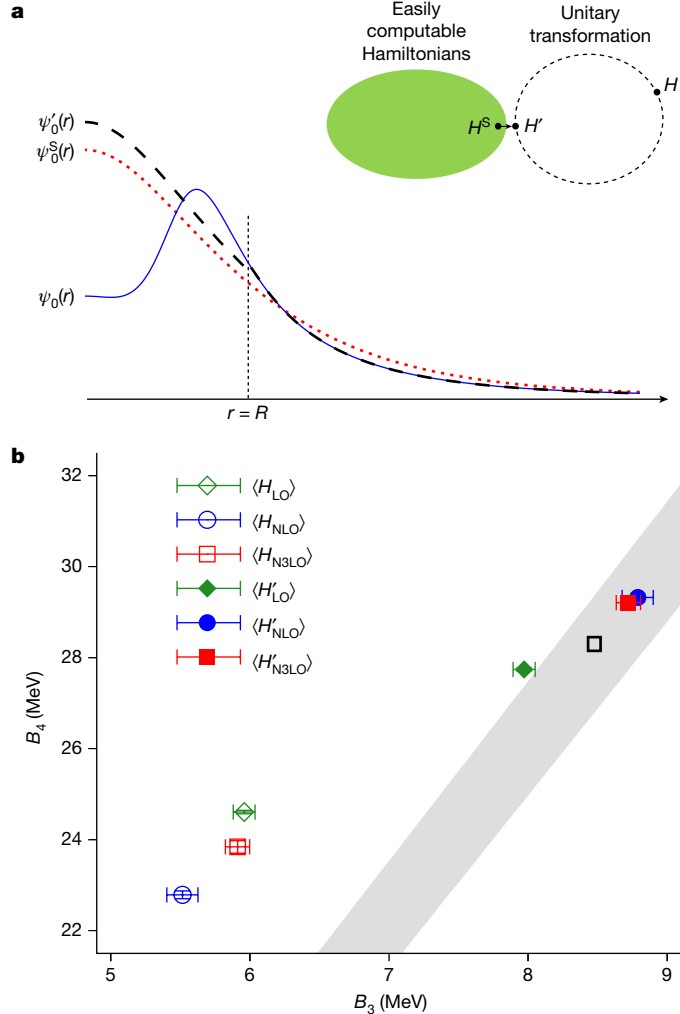

**Fig. 1 | Wavefunction matching and the Tjon band. a**, Pictorial representation of wavefunction matching. The simple Hamiltonian $H^S$ is an easily computable Hamiltonian whereas the high-fidelity Hamiltonian $H$ is not. A unitary transformation on the two-nucleon interaction with finite range $R$ is used to produce a new Hamiltonian $H'$ that is close to $H^S$. In each two-body channel, the ground-state wavefunction of $H'$ matches the ground-state wavefunction of $H$ for $r > R$ and is proportional to the ground-state wavefunction of $H^S$ for $r < R$. **b**, The Tjon band correlation between the binding energies of $^3$H ($B_3$) and $^4$He ($B_4$). The grey band is the predicted result from ref. 31. The black open box shows the empirical point. The green diamond, blue circle and red square points show the results at LO, NLO and N3LO in chiral effective field theory, respectively. The open points show the results from the first-order perturbative calculations using the Hamiltonian $H$ and the filled points are the results of the first-order perturbative calculations using the Hamiltonian $H'$. The error bars show standard deviations.

Hamiltonian is chosen so that the constant of proportionality is close to 1. For $r > R$, however, $U$ is not active and so $\psi'_0(r)$ remains equal to $\psi_0(r)$. The key point to notice here is that $\psi'_0(r)$ and $\psi^S_0(r)$ are numerically close to each other for all values of $r$. This can be seen visually in Fig. 1a and is the reason why perturbation theory in powers of $H' - H^S$ converges quickly when starting from low-energy states of $H^S$.

Wavefunction matching will now be applied to ab initio Monte Carlo nuclear lattice simulations[15,16,25,26,28] using the framework of chiral effective field theory (χEFT)[17,29]. For our realistic Hamiltonian $H$, we use χEFT two-nucleon interactions at next-to-next-to-next-to-leading order (N3LO) with lattice spacing $a = 1.32$ fm using a low-energy scheme described in Supplementary Information. For our simple Hamiltonian $H^S$, we use a χEFT interaction at leading order. Details of the interactions

can be found in Supplementary Information. In the following, we use the term 'local' for interactions that do not change the positions of particles and 'non-local' refers to interactions that do change the relative positions of particles. The 'range' of the interaction refers to the separation distance beyond which the interaction between particles becomes negligible.

We calculate all quantities up to first order in perturbation theory, which corresponds to one power in the difference $H' - H^S$. As a first test, we consider the energy of the deuteron, $^2$H. The wavefunction-matching calculation gives a binding energy of 2.02 MeV, compared with 2.21 MeV for the true binding energy of $H$ and 2.22 MeV for the experimentally observed value. The residual error of 0.1 MeV per nucleon is due to corrections beyond first order in powers of $H' - H^S$. If one does not use wavefunction matching and instead performs the analogous calculation to first order in $H - H^S$, the result is a much less accurate binding energy of 0.68 MeV.

As a second test of wavefunction matching, we calculate the binding energies of $^3$H and $^4$He. The Tjon band describes the universal correlations between the $^3$H and $^4$He binding energies[30,31]. Provided that there are no long-range non-local interactions, any realistic two-nucleon interaction produces binding energies that lie on the Tjon band. The inclusion of any short-range three-nucleon interaction also preserves this universal relation. In Fig. 1, we show wavefunction-matching calculations using two-nucleon interactions only. At leading order (LO) the calculated point falls outside the Tjon band as the Coulomb interaction is not included, whereas the next-to-leading order (NLO) and N3LO results lie squarely in the middle of the band. We are using a low-energy scheme where the two-nucleon interaction is the same at NLO and next-to-next-to-leading order (NNLO)[32]. The empirical point is also shown in Fig. 1. The good agreement with the Tjon band suggests a residual error of 0.1 MeV per nucleon or less for $^3$H and $^4$He. In Supplementary Information, we present numerical evidence that the estimate of 0.1 MeV error per nucleon is also valid for light and medium-mass nuclei. This can be compared with the substantial deviation from the Tjon line if one does not use wavefunction matching and performs the analogous calculation to first order in $H - H^S$. Before proceeding to larger nuclei and many-body systems, we first comment on the current status of ab initio calculations of nuclear structure using χEFT. The following analysis is not directly connected to wavefunction matching. Instead, it is a separate theoretical framework designed to help push beyond the current limitations of ab initio nuclear structure theory.

There has been tremendous progress in the past few years towards producing accurate results for nuclear structure across much of the nuclear chart using a variety of different computational approaches[33–44]. But there is also ample evidence that the calculations are sensitive to the manner in which the short-distance features of the interactions are regulated[20,45–48], a warning sign that systematic errors are not fully under control. Current ab initio calculations have difficulty simultaneously maintaining high-fidelity two-nucleon phase shifts and mixing angles and describing the saturation energy and density of symmetric nuclear matter as well as the binding energies and charge radii of light and medium-mass nuclei. Previous ab initio nuclear structure calculations have either not addressed some of the relevant observables or require further improvement in one or more of these areas. We aim to identify the problem and point to a viable solution.

The results in refs. 49,50 showed that the range and locality of the nuclear interactions have a strong influence on nuclear binding and that the α–α interaction is highly sensitive to the range and locality of the nucleonic interactions as well as omitted higher-order interactions. These same arguments apply to other interactions involving α particles and nucleons. In Supplementary Information, we use the formalism of cluster effective field theory[51–54] for α-particles and nucleons to provide a simple counting argument for the number of parameters that require tuning to reduce unwanted errors. Our strategy is to tune the short-distance features of the three-nucleon interactions to achieve

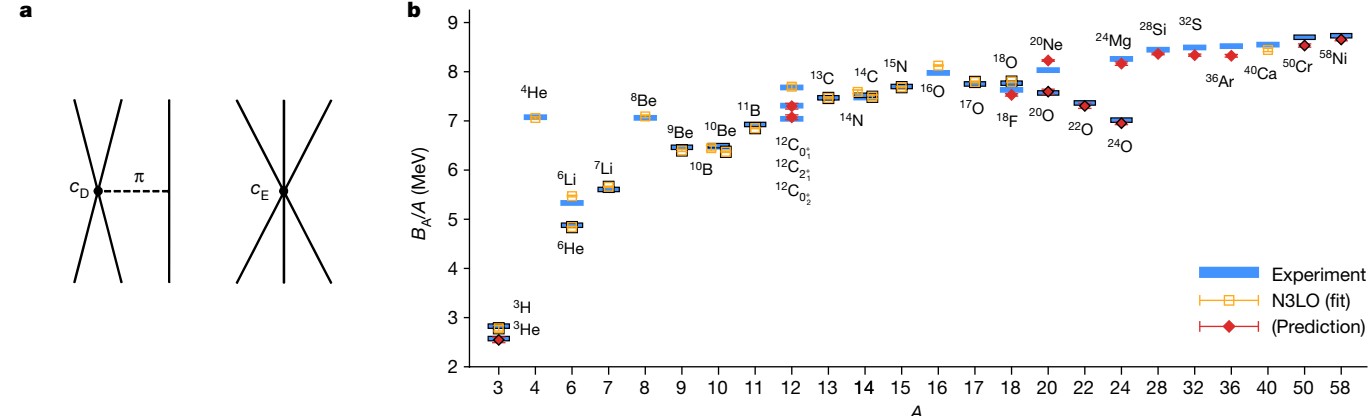

**Fig. 2 | Short-range three-nucleon forces at NNLO and results for nuclear binding energies. a**, Short-range three-nucleon forces at NNLO. The first is the one-pion exchange term $c_D$ shown on the left. The other is the purely short-range term $c_E$ shown on the right. At order N3LO, there are additional three-nucleon interactions associated with the exchange of two pions, as well as the corrections from the renormalization of the $c_D$ and $c_E$ terms. **b**, Results for nuclear binding energies ($B_A$) using wavefunction matching. Calculated ground-state and excited-state energies of some selected nuclei with up to $A = 58$ at N3LO in χEFT and comparison with experimental data. The symbols with a black border indicate nuclei with unequal numbers of protons and neutrons. The nuclei used in the fit of the higher-order three-nucleon interactions are labelled with open squares and the other nuclei are predictions denoted with filled diamonds. The error bars show standard deviations.

this error cancellation. We should emphasize that our calculations are full $A$-body calculations, and cluster effective field theory is only used to diagnose sensitivities to short-distance physics.

In χEFT, three-nucleon forces first appear at order NNLO. These include terms associated with the exchange of two pions and whose coefficients are determined from pion–nucleon scattering. There are also two interactions with singular short-distance properties that must be regulated and the corresponding couplings fitted to empirical data. As shown in Fig. 2a, $c_D$ corresponds to the short-range interaction of two nucleons linked to a third nucleon through the exchange of a pion, and $c_E$ corresponds to the short-range interaction of all three nucleons. At N3LO, there are additional terms associated with the exchange of two pions as well as readjustments of the $c_D$ and $c_E$ coefficients[55–57]. Four-nucleon interactions also appear at N3LO but are not considered in this work.

We tune the short-distance features of the $c_D$ and $c_E$ three-nucleon interactions to minimize errors in the binding energies of selected light and medium-mass nuclei. A total of six additional three-nucleon parameters are adjusted, and in Supplementary Information we present the details of these parameters along with a detailed description of the fitting procedure and the resulting uncertainty. We find that with just one parameter, the root-mean-square-deviation (RMSD) for the energy per nucleon drops from 1.2 MeV down to 0.4 MeV. With the addition of a few additional parameters, the RMSD per nucleon drops further to about 0.1 MeV. These results are consistent with the hypothesis that the α–α interaction has a key role in nuclear binding and that there are several additional cluster interactions that are sensitive to short-distance physics.

In Fig. 2b, we present the results for the nuclear binding energies using wavefunction matching. We show ground-state and excited-state energies of selected nuclei with up to $A = 58$ nucleons and comparison with experimental data. The symbols with a black border indicate nuclei with unequal numbers of protons and neutrons. The nuclei used in the fit of the three-nucleon interactions are labelled with open squares, and the other nuclei are predictions denoted with filled diamonds. The one-standard-deviation error bars shown in Fig. 2 represent uncertainties due to Monte Carlo errors, infinite-volume extrapolations and infinite projection time extrapolations. As described in Supplementary Information, we estimate the additional systematic errors due to truncation of the expansion in powers of $H' - H^s$ to be approximately 0.1 MeV per nucleon. However, this source of systematic error can be

significantly reduced by allowing for variational optimization of the Hamiltonian used to prepare the nuclear many-body wavefunction. We perform this variational optimization so that the remaining systematic error is smaller than the estimated computational error due to other sources. In Supplementary Information, we also compute the additional systematic errors due to uncertainties in the chiral interactions.

In Fig. 3a, we present the results for the charge radii of nuclei with up to $A = 58$ nucleons. No charge radii data were used to fit any interaction parameters. The one-standard-deviation point estimate error bars shown in Fig. 3 represent computational uncertainties due to Monte Carlo errors, infinite-volume extrapolation and infinite-time extrapolation. The agreement with empirical results is quite good, with an RMSD of about 0.03 fm. An extended analysis for selected nuclei that also includes uncertainties from the interactions are presented in Supplementary Information. We note that the larger errors for the heaviest nuclei are statistical and can be decreased by utilizing greater computational resources. The specific terms included in the calculations of the charge radii are detailed in Supplementary Information.

In Fig. 3b, we present lattice results for the energy per nucleon versus density for pure neutron matter and symmetric nuclear matter. None of the neutron-matter and symmetric nuclear-matter data were used to fit any interaction parameters. The density is expressed as a fraction of the saturation density for nuclear matter, $\rho_0 = 0.16$ fm$^{-3}$. For the neutron-matter calculations, we consider 14 to 80 neutrons in periodic box lengths ranging from 6.58 fm to 13.2 fm. For the symmetric nuclear-matter calculations, we use system sizes from 12 to 160 nucleons in a periodic box of length 9.21 fm. The comparisons with several other published works are shown and detailed in the figure caption. We see that the neutron-matter calculations agree well with previous calculations. Within the uncertainties due to finite system size corrections, the symmetric nuclear-matter calculations show saturation at an energy and density consistent with the empirical saturation point labelled with the black rectangular box. The relative uncertainties due to finite system size are at the 10% level for the energy. Additional calculations with larger systems are needed to reduce the thermodynamic extrapolation error further.

The one-standard-deviation point estimate error bars shown represent computational uncertainties due to Monte Carlo errors and infinite projection time extrapolation. These lattice simulations of symmetric nuclear matter are qualitatively different to other theoretical calculations that assume a homogeneous phase. The lattice

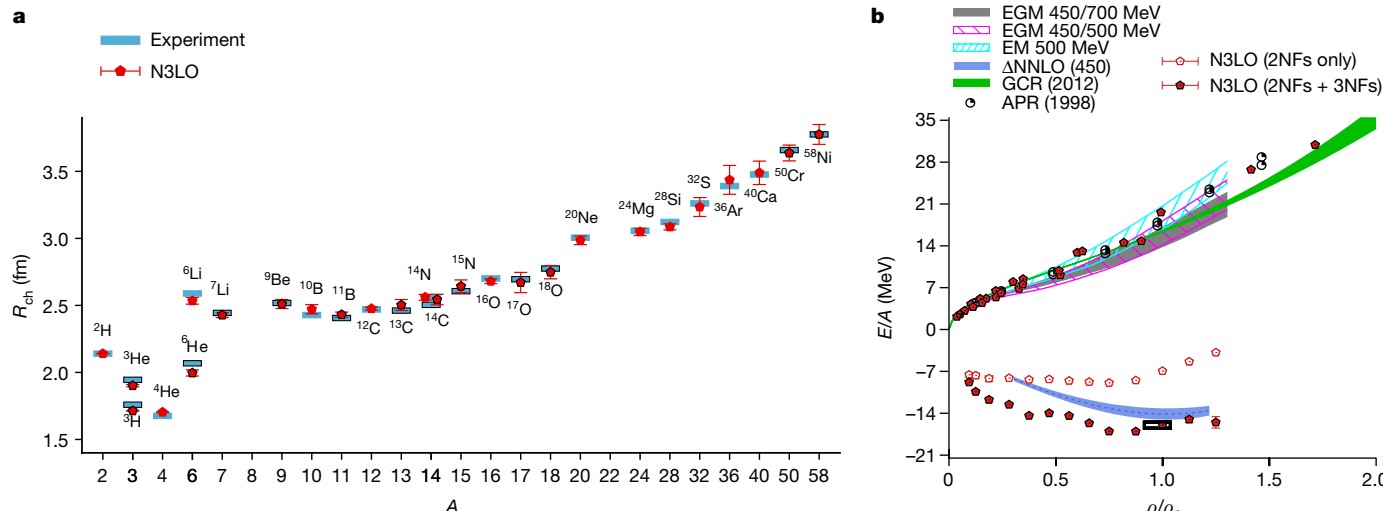

**Fig. 3 | Predictions for charge radii of nuclei and for pure neutron-matter energy per neutron and symmetric nuclear-matter energy per nucleon.**
**a**, Predictions for charge radii ($R_{ch}$) of nuclei up to $A = 58$ at N3LO in χEFT and comparison with experimental data. The symbols with a black border indicate nuclei with unequal numbers of protons and neutrons. **b**, Predictions for pure neutron-matter energy per neutron and symmetric nuclear-matter energy per nucleon as a function of density at N3LO in χEFT. For pure neutron matter, we use the number of neutrons from 14 to 80 and various box sizes from 6.58 fm to 13.2 fm. For symmetric nuclear matter, we use nucleon numbers from 12 to 160

and a periodic box of length 9.21 fm. For comparison, we show the results from variational calculations (APR)[65], auxiliary-field diffusion Monte Carlo simulations (GCR)[66], many-body perturbation theory using N3LO/NNLO (two-nucleon (2NF)/three-nucleon (3NF)) chiral interactions (EM 500 MeV, EGM 450/500 MeV and EGM 450/700 MeV)[67] and coupled cluster theory using NNLO chiral interactions with explicit delta degrees of freedom (ΔNNLO)[68]. The empirical saturation point is labelled with a black rectangular box. $E$ denotes energy, $\rho$ is the nucleon density, and $\rho_0$ is the saturation density of symmetric nuclear matter. The error bars show standard deviations.

simulations show phase separation and cluster formation, just as in the real physical system. Owing to the finite number of nucleons in these calculations, some oscillations due to nuclear shell effects can be seen in the energy per nucleon.

Another interesting feature of the lattice results is that symmetric nuclear matter without three-nucleon forces is underbound rather than overbound. This is different from what is found in other calculations using renormalization-group methods[58–60]. As discussed in Supplementary Information, wavefunction matching is very different from renormalization-group transformations. Wavefunction matching implements a unitary transformation that has finite range, and the process can be viewed as defining a new χEFT two-nucleon Hamiltonian $H'$. The interaction in $H'$ has a range no larger than that of $H$ and $H^S$ for the low-energy interactions. Therefore, one does not need to reconstruct the many-body forces induced by the unitary transformation and can simply treat $H'$ as the new χEFT two-nucleon Hamiltonian. Wavefunction matching has some characteristics similar to the unitary correlation operator method (UCOM)[61–63]. However, the unitary transformation in UCOM has properties that are more similar to renormalization-group transformations and, therefore, is also quite different from wavefunction matching. The induced forces generated by wavefunction matching have been investigated in a toy model[64]. A detailed discussion of the theory and applications of wavefunction matching and its implementation in continuous space are presented in Supplementary Information.

In summary, we have presented an approach for solving quantum many-body systems called wavefunction matching. Wavefunction matching uses a transformation of the particle interactions to allow for calculations of systems that would otherwise be difficult or impossible. We have applied the method to lattice Monte Carlo simulations of light nuclei, medium-mass nuclei, neutron matter and nuclear matter using high-fidelity chiral interactions and found good agreement with empirical data. Judging from the accuracy of the predictions, we have been successful in cancelling systematic errors in nuclear structure calculations by tuning the short-distance features of the three-nucleon

interactions. These developments may help resolve long-standing challenges in ab initio nuclear structure theory.

Although we have focused on Monte Carlo simulations for nuclear physics here, wavefunction matching can be used with any computational method and applied to any quantum many-body system. This also includes quantum computing algorithms where wavefunction matching can be used to reduce the number of quantum gates required. All that is needed is a simple Hamiltonian $H^S$ that produces fair agreement with empirical data for the many-body system of interest and is easily computable using the method of choice. Further details on the implementation and theory of wavefunction matching are given in Supplementary Information.

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

# Article

## Data availability

All of the data produced in association with this work have been stored and are publicly available at https://drive.google.com/drive/folders/1MByuG6NMagcgmURe4py-kwr9vksnHCl4.

## Code availability

All of the codes produced in association with this work have been stored and can be obtained upon request from the corresponding author, subject to possible export control constraints.

**Acknowledgements** We thank members and partners of the Nuclear Lattice Effective Field Theory Collaboration (J. Drut, G. Jansen, S. Krieg, Z. Ren, A. Sarkar and Q. Wang) and S. Bogner, A. Ekström, H. Hergert, M. Hjorth-Jensen, D. Phillips, A. Schwenk and W. Nazarewicz for discussions. We acknowledge funding by the Deutsche Forschungsgemeinschaft (DFG, German Research Foundation) and the NSFC through the funds provided to the Sino-German Collaborative Research Center TRR110 'Symmetries and the Emergence of Structure in QCD' (DFG project ID 196253076 - TRR 110, NSFC grant number 12070131001), the Chinese Academy of Sciences (CAS) President's International Fellowship Initiative (PIFI) (grant number 2018DM0034), Volkswagen Stiftung (grant number 93562), the European Research Council (ERC) under the European Union's Horizon 2020 research and innovation programme (ERC AdG EXOTIC, grant agreement number 101018170, and ERC AdG NuclearTheory, grant agreement number 885150), the Scientific and Technological Research Council of Turkey (TUBITAK project number 120F341), the National Natural Science Foundation of China (grants numbers 12105106 and 12275259), NSAF No.U2330401, US National Science Foundation (PHY-1913620, PHY-2209184, PHY-2310620), US Department of Energy (DE-SC0021152, DE-SC0013365, DE-SC0023658, DE-SC0024586, NUCLEI SciDAC-5 project DE-SC0023175), the Rare Isotope Science Project of the Institute for Basic Science funded by the Ministry of Science and ICT (MSICT), the National Research Foundation of Korea (2013M7A1A1075764, RS-2022-00165168), the Institute for Basic Science (IBS-R031-D1, IBS-I001-D1) and the Espace de Structure et de réactions Nucléaires Théorique (ESNT) of the CEA DSM/DAM. Computational resources provided by the Gauss Centre for Supercomputing e.V. (www.gauss-centre.eu) for computing time on the GCS Supercomputer JUWELS at Jülich Supercomputing Centre (JSC) and special GPU time allocated on JURECA-DC as well as the Oak Ridge Leadership Computing Facility through the INCITE award 'Ab-initio nuclear structure and nuclear reactions', and partially provided by TUBITAK ULAKBIM High Performance and Grid Computing Center (TRUBA resources). Computational resources were also partly provided by the National Supercomputing Center of Korea with supercomputing resources including technical support (KSC-2021-CRE-0429, KSC-2022-CHA-0003, KSC-2023-CHA-0005), and the Southern Nuclear Science Computing Center in the South China Normal University. We have complied with community standards for authorship and all relevant recommendations with regard to inclusion and ethics.

**Author contributions** Code development, testing, optimization and production runs were led by S.E. with contributions from F.H., M.K., T.A.L., D.L., N.L., B.-N.L., Y.-Z.M., G.R., S.S. and Y.-H.S. Conceptual work and tests were led by L.B. with contributions from S.E., E.E., D.F., H.K. and D.L. S.E., T.L., D.L., U.-G.M. and Y.K. supervised the research effort. Additional code development, testing, optimization and production runs in response to reviewer comments were performed by Y.-Z.M. The literature search was led by G.S. All authors contributed to the writing, editing and review of this work.

**Competing interests** The authors declare no competing interests.

### Additional information
**Correspondence and requests for materials** should be addressed to Dean Lee.
