## [Peer Review File · Nature]

Manuscript Title: Wave function matching transformations for solving the quantum many-body problem

Reviewer Comments & Author Rebuttals

Reviewer Reports on the Initial Version:

Referees' comments:

Referee #1 (Remarks to the Author):

A. Summary of the key results

The authors report on accurate Quantum Monte-Carlo computations of nuclear binding energies and charge radii of atomic nuclei for selected nuclei from the deuteron up to calcium-40. They also compute infinite neutron and symmetric nuclear matter, and accurately compute the nuclear matter saturation point. They employ Nuclear Lattice Effective Field Theory (NLEFT) and a novel technique called wave-function matching. This method alleviates the fermion sign problem, and thereby addresses a long-standing problem in Quantum Monte-Carlo simulations.

B. Originality and significance: if not novel, please include reference

The wave-function matching technique is an original and novel idea. However, the authors claim: "...no ab initio calculations have been able to provide an accurate description of nuclear binding energies, charge radii, and the saturation properties of symmetric nuclear matter with equal numbers of protons and neutrons. Our goal here is to identify the problem and point to a viable solution." This is not really a correct statement. References [19, 56] and [A. Ekström et al., Phys. Rev. C 91, 051301 (2015); C. Drischler et al., Phys. Rev. Lett. 122, 042501 (2019); V. Somà et al., Phys. Rev. C 101, 014318 (2020)] are arguably as accurate as the results in the present manuscript. It is impressive that accurate radii came out of the calculations without being included in the fit.

C. Data & methodology: validity of approach, quality of data, quality of presentation

Criticism to the methodology:

1. The authors write in the abstract that they use interactions from chiral effective field theory (EFT) at next-to-next-to-next-to-leading order (N³LO). This is not correct. They omit four-body forces that enter at this order, and they also omit long-range three-nucleon forces at N³LO. Furthermore, they employ eight three-body contacts instead of only two that exist in chiral EFT.
2. The authors write that the wave-function matching happens at a short distance R . However, they choose $R = 3.7\text{fm}$, almost three lattice spacings (1.32fm) and significantly larger than the nuclear effective range. Should R not be on the order of the lattice spacing? How sensitive are the results to the

variation of R ?

3. The “short-range” three-body contact from equation S21 couples nucleons that are two lattice spacings apart. This does not seem to be a short-range force. This term is quite important for saturation, and deviates from chiral EFT.

4. Are the values for the six additional three-nucleon low-energy constants natural in size?

5. The approach seems to have little predictive power. 17 nuclei, including calcium-40, were included in the fit of the three-nucleon force.

6. It is not clear from Fig. 3 that nuclear matter saturates at an accurate point.

D. Appropriate use of statistics and treatment of uncertainties

The authors use state-of-the-art tools such as emulators and history matching to arrive at their results. They only give statistical uncertainties on their main results. One wonders what the sensitivities are with respect to the three-body parameters and the matching distance R ?

E. Conclusions: robustness, validity, reliability

The abstract contains incorrect or misleading statements (see point A above). The results certainly seem valid, but they are not in the framework of chiral EFT.

F. Suggested improvements: experiments, data for possible revision

1. Addressing the questions raised under points C and D.

2. Include the formula used to compute the charge radii. Did the authors include the spin-orbit correction?

G. References: appropriate credit to previous work?

Missing references are listed under point B.

H. Clarity and context: lucidity of abstract/summary, appropriateness of abstract, introduction and conclusions

The paper is well written, with the caveats presented in points B and D.

Referee #2 (Remarks to the Author):

Dear Editor,

in the manuscript « Wave function matching for the quantum many-body problem », the authors,

Elhatisari et al., present a new method to compute the structure of many-body quantal systems. In particular, they look at light- to medium-mass nuclei. These structures are difficult to compute because of the complexity of the interaction between the nucleons. Finding the eigenstates of modern, realistic nuclear Hamiltonians is therefore quite a difficult task. The idea behind the « wave-function matching » method is to transform the realistic Hamiltonian H into another Hamiltonian H' through a unitary transformation U . That new Hamiltonian therefore exhibits the exact same physics content as the initial H , but it is chosen so that the short-range behaviour of its wave functions is proportional to that of a simple many-body Hamiltonian $H^{\wedge S}$ of which the eigenstates can be easily computed. The particular unitary transformation U that enables such properties of the eigenstate wave functions of H' is the Hamiltonian translator (S1) applied between $H^{\wedge S}$ and H .

The present manuscript shows the results obtained with this new, innovative, method on light- to medium-mass nuclei computed with Monte-Carlo Quantum calculations. Excellent agreement is obtained with experiment, even on nuclei not used to fit the low-energy constants of the nuclear interaction considered by the authors. This suggests the method has a good predictive power. Unfortunately, the manuscript lacks clarity and it is not always clear what has been done. I understand that within a short text, it is not possible to include all the necessary details to be able to reproduce the authors' calculations. Nevertheless, that is the purpose of the « Methods » section. The authors should address the following issues prior to a possible acceptance of their manuscript.

My major concern is on the test of the method. Would it be possible to confront this new method to other many-body techniques using one given nuclear interaction? Such a test could be performed on a small system like ${}^4\text{He}$, which is accessible to various exact methods. The authors seem to make this test, but only on the deuteron, of which the structure is not affected by three-body forces, and the test illustrated in the right panel of Fig. 1 on the triton and ${}^4\text{He}$ does not include three-body forces. Accordingly, the N3LO calculations (solid symbols) does not reproduce the data (black open box). To demonstrate that this new method works, a fully converged calculation using the wave function matching should be compared to another « exact » method using exactly the same nuclear interaction. I guess that a comparison with the no-core shell model or the Green's function Monte Carlo method could be done using the N3LO chiral EFT interaction considered by the authors. Can this test be performed? In the affirmative, it should be shown to prove the validity of the approach. If not, it would be difficult to accept this as a reliable way to compute many-body quantal systems.

Related to the previous issue, as in other nuclear-structure techniques, the wave function matching induces many-body forces (see the one-before-last paragraph on p. 4). This leads the authors to determine new coefficients for, e.g., three-body forces, see p. 11 in the « Methods » section. If these parameters have to be re-fitted, what is the actual predictive power of the wave function matching? Do I understand correctly that a calculation using, e.g., a given chiral EFT nuclear interaction with three-body forces, would not lead to accurate binding energies for $A > 2$ nuclei? Accordingly, could these calculations really be called « ab initio »?

The implementation of the method is also not very clear. The authors mention an « expansion in powers

of the difference $H' - H^{\wedge}S$ » (see the bottom of p. 1). Does this mean that the translator (S1) is expanded in powers of H_T , which is a difference of two Hamiltonians? If yes, wouldn't a Padé expansion be more appropriate to conserve the unitarity of the translator?

The calculations shown in the right panel of Fig. 1 are performed at the first order of that expansion. What about the results shown in Figs. 2 and 3? Are they also obtained at the sole first order of that expansion or were they obtained with higher orders of the expansion scheme?

On p. 3, the authors make a comment on cluster EFT (see the third paragraph). I do not really see how this discussion on the cluster EFT fits within this work. Were the calculations obtained assuming an alpha-cluster structure of the nuclei, similar to the Ikeda diagram (plus maybe a few nucleons when N does not equal Z)? [see Ikeda et al. Prog. Theor. Phys. Suppl. 464 (1968)] Or are the calculations shown here full A -body calculations, where A is the mass number?

In the subsection « Wave Function Matching » of the « Methods » section, the authors « restrict [themselves] on the space of two nucleons ». What is done in actual calculations? Is the wave-function-matching method applied only to the two-body part of the nuclear interaction, or does it affect also the many-body wave function?

In addition to these main questions, I have other comments on the manuscript; there remain some typos and some sentences could be improved to ease the reading of the text:

- p. 2, end of the paragraph below Fig. 1: « the separation distance beyond which » (replace « at » by « beyond »?)
- p. 2, last line: « (NLO). » (remove one of the two full stops).
- p. 3, middle of the page: « At order N3LO, there are additional terms ... at N3LO order »; one of the « N3LO » should be removed. Note also that O in N3LO means « order », so « N3LO order » is redundant. Same comment in the caption of Fig. 2.
- To what does the black open box correspond in the right panel of Fig. 3?
- p. 7, 3rd paragraph, after its definition, the function « $f(t)$ » has an argument t/T , which is not consistent. Actually, « $f(t)$ » is the value of the function in t , so writing « Let f be a smooth function... » would be enough.
- In the subsection « Wave Function Matching », it would help to relate H , $H^{\wedge}S$ and H' to the Hamiltonians H_A , H_B , $H_{A'}$ and $H_{B'}$ of the previous subsection « Hamiltonian Translators ». This would help the reader understand what is actually done in this new method. If the authors develop (S1) in powers of H_T , this would also be good place to mention it.
- p. 9, first line below Eq. (S17): K has already been defined on the previous page [see below Eq. (S4)].
- p. 10, the authors mention for the first time the « history matching ». Unfortunately, this method is defined only at the bottom of p. 11, and the corresponding Ref. 60 is provided only on p. 13. This should be revised.
- p. 10, two lines above Eq. (S26): « $f(x)$ » is mentioned but Eq. (S26) define f_o . Is this a typo?
- p. 11, first paragraph: to what do « $V^{\wedge}(0)_{cE}$, $V^{\wedge}(1)_{cE}$, $V^{\wedge}(2)_{cE}$, $V^{\wedge}(0)_{cD}$, $V^{\wedge}(1)_{cD}$, and $V^{\wedge}(2)_{cD}$ » correspond? Are they related to the expressions detailed in Eqs. (S20), (S21), and (S22), or do

they correspond to other three-body forces? In the latter case, wouldn't it be useful to include the expression and physical meaning of these forces? Note that a short physical explanation of the difference between $V^{\text{(l)}}_{\text{cE}}$, $V^{\text{(t)}}_{\text{cE}}$ etc. would be helpful.

- p. 14, within the paragraph « WAVE 3 », there are weird spaces after « WAVE 2 » and « WAVE 3 » within the text.

- p. 14, third line of the paragraph « WAVE 3 »: « we run the implausibility measure » (one « the » too many).

- p. 16, third line: « clearly indicate » should be at the plural form because the subject is « The first few rows ».

In conclusion, this manuscript presents the wave function matching, a new method to compute accurately many-body quantal systems. Its application is illustrated here on the complex and difficult problem of atomic nuclei. However, if proved exact, it could probably be very useful beyond the domain of nuclear physics, for example in atomic or solid-state physics. Unfortunately, and probably because of the short length of the manuscript, the details on the method and its actual implementation are not very clear. The authors should address the different issues raised in this report to make their manuscript accessible to the broad readership of Nature.

Referee #3 (Remarks to the Author):

The Authors present a new method called wave function matching that is designed to quench sign problems in lattice Monte Carlo calculations. They use this method with high-fidelity chiral EFT interactions and lattice EFT to compute energies and radii in finite nuclei and the equation of state for infinite nuclear & neutron matter. The Authors claim to provide some insights that may help resolve long-standing challenges in accurately reproducing bulk properties of nuclei and saturation of nuclear matter.

On the one hand, the method of wave function matching, accompanied by six additional and non-standard short-range terms in chiral EFT, is original and very creative. The fact that this approach solves the problem of sign cancellations of high-fidelity Hamiltonians on the lattice and predict good radii, once the short-range terms are fitted to energies in heavy systems, is certainly an accomplishment. The manuscript presents technical developments that I believe is of interest to the broader physics community as well as developers & practitioners of Monte Carlo methods.

On the other hand, I am unsure whether this technical development warrants publication in Nature and I have a series of concerns that

cast doubt on the physics conclusions that can be drawn from the presented results. In particular regarding the predictive power, physics insight, and systematic improvement once the additional short-range terms have been introduced. Do these terms solve a technical Monte Carlo problem or a physics problem? Can the Authors present evidence that clarifies this question and strengthens the physics conclusions in the manuscript?

Below, is a list of comments (not in order of relevance) that I believe should be addressed by the Authors.

1.) As the Authors point out, the long-standing challenge of accurately reproducing energies, radii, and saturation has received a lot of attention. It is still not clear what makes some interactions give reliable results for heavier systems and others not. This is an important question and we certainly need a much better understanding of the nuclear interaction, and new takes on the problem are important.

However, there are results (e.g. Refs. 19,56) based on NN+3NF interactions in chiral EFT up to third order (NNLO) that describe energies and radii of selected nuclei up to 208Pb and nuclear matter saturation consistent with experimental data, without the need for additional short-range terms. Those calculations also employ a regulator cutoff similar to the one imposed by the lattice EFT spacing in the present work. The fourth order (N3LO) corrections, and somehow beyond, as is done in the manuscript, should slightly improve predictions. Judging from Fig. S4 there appears to be a nearly 50% correction in the 40Ca energy coming from the higher order short-range terms. Why not re-fit the standard 3N terms c_D and c_E to some heavy data in wave 1? What are the new insights and what is gained by adding more 3N contacts? Does the approach qualify as an EFT and how to systematically improve?

2.) The Authors utilize the 3N short-range terms to remove unwanted dependence on short-range physics. However, the matching radius R is set to 3.7 fm, which is nearly three times the lattice spacing and on good grounds cannot be considered short range. Even 1.9 fm, which is also quoted, is of rather long range. How can this value of R be understood as short range?

3.) For relative distances $r > 3.7$ fm, the unitary transformation U is

not active so the Hamiltonian and the 2N wave functions are given by their high-fidelity versions, which at this range should be dominated by the leading-order one-pion exchange. For $r < 3.7$ fm, U is active and after matching the 2N wave functions are proportional, with a constant ~ 1 , to the leading-order and simple wave functions. A set of short-range 3N terms are then added and re-fitted to energies in heavy nuclei. This makes me wonder how different the results would be if one used only the leading-order 2N interaction plus a few (up to six or eight?) short-range 3N terms fitted to heavier systems instead of wave function matching?

Related to this, there are also Lattice EFT calculations by Lu et al. (Physics Letters B 797 (2019) 134863) based pionless EFT NN+3N potentials with only 4 parameters, of which 2 are contacts, that yield results very similar to the ones in the present manuscript.

4.) As described in the Methods section, the Authors have carried out a careful history matching procedure complemented with a least-squares minimization to identify non-implausible and optimal LEC values. However, the number of unknown 3N short-range terms are comparable or even equal to the number of nuclear energies used for calibration in some waves of the history matching. Accounting for the fact that the leading-order couplings $c_{SU(4)}$ and c_I are allowed to vary appears to yield more parameters than data in waves no. 1/2. The situation is better in waves no. 4/5. Is this a problem? Are the Authors concerned about over-fitting in the least squares?

5.) Figs. 2 and S4 nicely demonstrates the well-known (Phys. Rev. C 64, 014001, and Ref. 81) improvement of energy predictions for $A > 4$ nuclei when including some of them in the fit. In Fig. 2, the predicted nuclei with masses in $A=3-40$ are in essence interpolated since the interaction is anchored at $A=3$ and $A=40$ and is calibrated in between. It would add some credibility to the results if the Authors included energy (and radii) predictions beyond ^{40}Ca using the same interaction.

6.) Fig. 3 shows predicted radii for $A=2-40$ and this is a very nice result. How well are the radii predicted for the other combinations of 3N contacts and interactions emerging from waves 1-5?

- Besides the comments above, the abstract plus main text is well written and so are many parts of the Methods section. In some places

clarification is needed and below is a list of minor comments to hopefully improve the manuscript.

A.) The value of the matching radius $R=3.7$ fm should be mentioned in the main text.

B.) Regarding the Nature Physics paper in Ref. 56; I consider this relevant also to the underlying physics discussion of the paper and not only the history matching part.

C.) It is not clear in the main text, or captions to Figs. 2 and 3, exactly which set of additional 3NF contacts are included and which of the many interactions listed in the Methods section that is used in the end. It can probably be deciphered from the Methods section, but it would help if this was pointed out in the main text.

D.) Are the results in Figs. 2 and 3. point-estimates or posterior predictive distributions? Also, it is not easy to find the lattice predictions for the pure neutron matter energies in Fig. 3.

E.) Systematic theoretical errors are estimated to be 0.1 MeV per nucleon. I could not find a clear explanation of this number in the Methods section.

F.) It would be useful to know how the numerical values of the smearing parameters below Eq. S7 are determined and whether this would impact the results?

G.) Maybe it is implicitly defined somewhere in the Methods section, but I could not see a clear definition of, or reference to, the modified $SU(4)$ short-range 3N terms. For instance, where is $V_{\{c_D\}^{(2)}}$ defined.

H.) The notation \mathbf{b}_x in connection with the emulator in Eq. S26 and minimization in Eq. S27 is confusing. Immediately below Eq. S26, the elements of this vector are declared to denote only(?) the NN LECs, i.e., the x_i 's in the first term of Eq. S26. After this, it becomes a bit muddy how the β_k 's enter the history matching analysis. For instance, it looks like only the NN LECs are fitted using least squares. Which I presume is not the case.

I.) How is the natural size for the 3N LECs β_k estimated? The

inference is conditional on this information and it should be quantified.

J.) Figure S3 should be equipped with numerical values on the x-axis to provide a scale for the posterior distribution of the NN LECs. The prior for the NN LECs should also be specified.

Author Rebuttals to Initial Comments:

General Comments

The authors thank all the referees for their insightful comments, questions, and suggestions. We have spent six months working to address all of the questions and suggestions thoroughly. We have expended great effort exploring all of the relevant issues in detail and improving the manuscript by including more context, insight, examples, and analysis. We are grateful to the editor and the referees for the valuable dialogue and opportunity to discover new science. During the six-month process, we have learned that wave function matching is theoretically more interesting and computationally more useful than we had realized when submitting the original manuscript.

In addition to addressing the topics raised by the referees, we have worked to increase the broad impact and lasting importance of this work by adding new material in Methods on the follow topics: (1) Demonstrating how wave function matching can be implemented in continuous space and so that it is immediately useful for other research groups using other *ab initio* methods and researchers beyond nuclear physics. (2) Showing how wave function matching can be used to construct new high-fidelity chiral interactions. (3) Proving that wave function matching does not change the asymptotic convergence properties of the chiral effective field theory expansion. (4) Explaining the importance of the $\alpha\alpha$ interaction for nuclear binding and presenting numerical evidence of its important role in quantifying regulator-dependent errors in many-body calculations.

Several research projects are now in progress that use the new methods and concepts developed in this paper. In Fig. P1 we show the ground-state binding energies per nucleon for the carbon and oxygen isotopes calculated by the NLEFT collaboration using wave function matching with chiral interactions at N3LO. This project is led by Myungkuk Kim, Young-Ho Song, and Youngman Kim. We note the excellent agreement with experimental data. In Fig. P2 we show the low-energy spectrum of the beryllium

[REDACTED]

isotopes calculated by the NLEFT collaboration using wave function matching with chiral interactions at N3LO. This new project is led by Shihang Shen. The agreement with experimental data using the wave function matching method for the N3LO interactions is outstanding. For comparison, we show results obtained using a simpler interaction (SU4) that is independent of spin and isospin. In Fig. P3 we show the charge radii of silicon isotopes calculated by the NLEFT collaboration using wave function matching with chiral interactions at N3LO. The new experimental results are obtained at the Facility for Rare Isotope Beams using laser spectroscopy. The lattice calculations are led by Yuanzhou Ma. Also shown are theoretical calculations obtained using valence-space in-medium similarity renormalization group with two different *ab initio* interactions. Of the five sets of theoretical results presented, only the

REDACTED

N³LO wave function matching lattice results accurately predict both the overall size of the charge radii and the isotopic dependence of the charge radii.

These results give concrete evidence that wave function matching for the nuclear many-body problem is on solid foundations. The new methods are computationally efficient, the results accurately describe experimental values, and the applications for new science discovery are now underway. Some additional applications of this work have already appeared in preprint form. The Coulomb force contributions to the ground state energies of various nuclei calculated here have been used in a novel and improved determination of the dependence of element generation in the Big Bang on the electromagnetic fine-structure constant α_{EM} [Meißner et. al, 2305.15849]. The interactions in this work have also been used to calculate the static structure factors for hot neutron matter [Ma et al., 2306.04500].

Reply to Reviewer 1

A. Summary of the key results

The authors report on accurate Quantum Monte-Carlo computations of nuclear binding energies and charge radii of atomic nuclei for selected nuclei from the deuteron up to calcium-40. They also compute infinite neutron and symmetric nuclear matter, and accurately compute the nuclear matter saturation point. They employ Nuclear Lattice Effective Field Theory (NLEFT) and a novel technique called wave-function matching. This method alleviates the fermion sign problem, and thereby addresses a long-standing problem in Quantum Monte-Carlo simulations.

B. Originality and significance: if not novel, please include reference

The wave-function matching technique is an original and novel idea. However, the authors claim: "... no ab initio calculations have been able to provide an accurate description of nuclear binding energies,

Figure P3. Preliminary results for the charge radii of silicon isotopes and comparison with new experimental data and other theoretical methods.

charge radii, and the saturation properties of symmetric nuclear matter with equal numbers of protons and neutrons. Our goal here is to identify the problem and point to a viable solution.” This is not really a correct statement. References [19, 56] and [A. Ekström et al., Phys. Rev. C 91, 051301 (2015); C. Drischler et al., Phys. Rev. Lett. 122, 042501 (2019); V. Somà et al., Phys. Rev. C 101, 014318 (2020)] are arguably as accurate as the results in the present manuscript. It is impressive that accurate radii came out of the calculations without being included in the fit.

We thank the referee for this correction. We have revised the statement and it now reads:

“Current *ab initio* calculations have difficulty simultaneously maintaining high-fidelity two-nucleon phase shifts and describing the saturation energy and density of symmetric nuclear matter as well as the binding energies and charge radii of light and medium-mass nuclei. Previous *ab initio* nuclear structure calculations have either not addressed some of the relevant observables or require further improvement in one or more of these areas. We aim to identify the problem and point to a viable solution.”

All of the works mentioned by the referee are excellent papers. However, the authors of each paper would agree with the following objective assessments of the areas that need improvement. In A. Ekström et al., Phys. Rev. C 91, 051301 (2015), the 3P_1 and 3P_2 phase shifts are not accurate at energies relevant to nuclear binding. Putting the phase shifts at their empirical values creates tension with other observables. As discussed in Nosyk et al., Phys. Rev. C 104, 054001 (2021), the 3S_1 and 1P_1 phase shifts in Jiang et al., Phys. Rev. C 102, 054301 (2020), are not accurate at energies relevant to nuclear binding, and putting the phase shifts at their empirical values create tension with other observables. Hu et al., Nature Phys. 18, 1196 (2022), uses the interactions developed in Jiang et al., Phys. Rev. C 102, 054301 (2020).

In C. Drischler et al., Phys. Rev. Lett. 122, 042501 (2019), calculations of binding energies and

charge radii of light and medium-mass nuclei are not presented. The calculations of binding energies and charge radii of medium-mass nuclei come later in J. Hoppe et al., Phys. Rev. C 100, 024318 (2019). There it is found that the binding energies are underestimated and the charge radii are overestimated. An alternate extension of the same interactions is considered in Hüther et al., Phys. Lett. B 808, 135651 (2020). There the binding energies and charge radii of medium-mass nuclei are fairly accurate, but the saturation properties of nuclear matter are significantly off.

In V. Somà et al., Phys. Rev. C 101, 014318 (2020), the binding energies of light and medium-mass nuclei are accurate. However, the charge radii are underestimated and the saturation properties of nuclear matter are not calculated.

C.:Data & methodology: validity of approach, quality of data, quality of presentation

Criticism to the methodology:

1. The authors write in the abstract that they use interactions from chiral effective field theory (EFT) at next-to-next-to-next-to-leading order (N3LO). This is not correct. They omit four-body forces that enter at this order, and they also omit long-range three-nucleon forces at N3LO. Furthermore, they employ eight three-body contacts instead of only two that exist in chiral EFT.

We thank the referee for this correction. In order to keep the abstract simple and understandable for nonspecialists, we now write:

“We use high-fidelity chiral effective field theory interactions and find good agreement with empirical data.”

In the main text we write:

“In χ EFT, three-nucleon forces first appear at order NNLO. These include terms associated with the exchange of two pions and whose coefficients are determined from pion-nucleon scattering. There are also two interactions with singular short-distance properties that must be regulated and the corresponding couplings fitted to empirical data. As shown in the left panel of Fig. 2, c_D corresponds to the short-range interaction of two nucleons linked to a third nucleon through the exchange of a pion, and c_E corresponds to the short-range interaction of all three nucleons. At next-to-next-to-next-to-leading order (N3LO), there are additional terms associated with the exchange of two pions as well as readjustments of the c_D and c_E coefficients [31-33]. In our work we do not include four-nucleon interactions which also appear at N3LO order.

Later we write:

“For our realistic Hamiltonian H , we use χ EFT two-nucleon interactions at N3LO with lattice spacing $a = 1.32$ fm using a low-energy scheme described in Methods.”

In Methods, we now write:

“As in previous studies using NLEFT, we use a low-energy scheme to simplify the operators obtained in

chiral effective field theory. The two-pion exchange interactions have an asymptotic spatial dependence proportional to $e^{-2M_\pi r}$ times power law factors of $1/r$. For our lattice spacing of $a = 1.32$ fm, the details of the two-pion exchange potential are not fully resolved. We therefore treat two-pion exchange interactions as well as higher-pion exchange interactions in the same manner as the short-range contact interactions. Within this low-energy scheme of chiral effective field theory, our lattice calculations include all two-nucleon and three-nucleon interactions up to $O(Q^4)$ or next-to-next-to-next-to-leading order (N3LO). The additional three-nucleon interactions used in our calculations correspond with a particular choice of the local regulators used for the three-nucleon interactions. We have not included four-nucleon interactions which also first appear at N3LO. In short, we are implementing chiral effective field theory at N3LO in the low-energy scheme with special local regulators for the three-nucleon interactions and without four-nucleon interactions.”

2. The authors write that the wave-function matching happens at a short distance R . However, they choose $R = 3.7$ fm, almost three lattice spacings (1.32 fm) and significantly larger than the nuclear effective range. Should R not be on the order of the lattice spacing? How sensitive are the results to the variation of R ?

We thank the referee for this important question. The short answer is that wave function matching with radius $R \approx 1.32$ fm would not be able to treat partial waves beyond the S-wave. For higher partial waves, the wave function vanishes at the origin. If there is only one allowed lattice radial distance for wave function matching, then the unitary transformation becomes trivial. With a smaller lattice spacing, one could perform wave function matching with $R \approx 1.32$ fm. However it is not clear that it would achieve the desired effect. In order to accelerate the convergence of perturbation theory, R needs to be somewhat larger than the range of the interactions in H and H^S . $R \approx 1.32$ fm would be too small. The concepts and reasoning are contained the material to follow.

In Methods we now have new section called “Dependence on the Wave Function Matching Radius R ”. In that section we write:

“For each scattering channel where wave function matching is used, we assume that E_0 and E_S are close in energy and the asymptotic normalization ratio κ is close to 1. These conditions are required for wave function matching to accelerate perturbation theory. They are satisfied for the applications of wave function matching to two-nucleon interactions in the main text, and they are also satisfied for the examples we discuss in the next few sections.

Let r_Δ be the largest radial distance for which the interactions comprising H and H^S are different and the difference is significant in magnitude. For the two-nucleon interactions in the main text, r_Δ is approximately one lattice spacing, or 1.32 fm. For the higher partial waves, the interactions vanish at zero distance and r_Δ extends somewhat further to $\sqrt{2}$ times the lattice spacing, or 1.86 fm. Since $E_0 \approx E_S$ and $\kappa \approx 1$, it follows that $\psi_0(r) \approx \psi_0^S(r)$ for all r greater than r_Δ .

The unitary transformation U used in wave function matching maps $|\psi_0^S\rangle_R$ to $|\psi_0\rangle_R$. Since the wave functions $|\psi_0^S\rangle_R$ and $|\psi_0\rangle_R$ are nearly equal for $r \geq r_\Delta$, the action of U on $|\psi_0^S\rangle_R$ can therefore be truncated to radial distances $r < r_\Delta$. For the transformation of $|\psi_0^S\rangle_R$, we conclude that there is no dependence on R once R is greater than r_Δ . On the other hand, U also transforms other two-body states at distances $r < R$. However, these states are orthogonal to $|\psi_0^S\rangle_R$. These states must have an extra node between $r = 0$ and

$r = R$ and therefore correspond to a relative momentum of size $O(2\pi/R)$ or larger. The dependence on R can only be seen in the interactions of the high-energy modes with momenta above $2\pi/R$. The low-energy physics of wave function matching is independent of R . While there are interactions in H' that reach up to radial distances of R , these interactions only couple to high-energy modes. This analysis shows that wave function matching is very different from renormalization group evolution. Changing R does not change the low-energy resolution scale, and H' behaves as a two-body Hamiltonian with interaction range set by r_Δ . This is in addition to any long-range interaction features such as the one-pion exchange potential common to both H and H^S and therefore trivially carried over to H' .

The value of $R = 3.72$ fm used in the main text corresponds to a momentum scale of $2\pi/R = 333$ MeV. Since 333 MeV is a high-momentum scale comparable to the cutoff scale of our low-energy effective field theory, we expect the low-energy physics is largely independent of R in this range. This is in fact what is observed. In Fig. S5 we plot the energies of ${}^3\text{H}$ and ${}^4\text{He}$ for different values of the wave function matching radius R . We show N3LO lattice results in chiral effective field theory for $R = 2.63$ fm, $R = 3.22$ fm, and $R = 3.97$ fm. The higher-order corrections are calculated using first-order perturbation theory. The gray band is the predicted result from Ref.,¹ and the black open box shows the empirical point.

We see that the results are largely independent of R . The slight deviation from the Tjon band for $R = 2.63$ fm is likely due to perturbation theory corrections beyond first order. The fast convergence of perturbation theory in wave function matching requires that the radius R is somewhat larger than r_Δ . This appears to be satisfied for $R = 3.22$ fm and $R = 3.97$ fm, and the results are nearly identical for the two values for R . They are also in excellent agreement with the N3LO results presented in the main text for $R = 3.72$ fm.”

In Methods, we have written a new section called “Induced Three-Body Interactions in Wave Function Matching: Born-Oppenheimer Analysis”. There we study the induced three-body interactions generated by wave function matching and the dependence on the wave function matching radius R . For this purpose, we use a Born-Oppenheimer analysis where two of the particles have infinite mass and the third particle has finite mass. For all possible separation distances between the two infinite-mass particles, we compare the ground state energies for the original Hamiltonian and the transformed Hamiltonian obtained using wave function matching with radius R . We find conclusive evidence that the low-energy physics is independent of the wave function matching radius R . We also conclude that the induced three-body interaction is numerically small.

In Methods, we have written another new section called “Induced Three-Body Interactions in Wave Function Matching: Three Particles in One Dimension”. There we study a three-body system composed of three distinguishable particles in one dimension. We compute the six lowest three-body energies in given a periodic box and compare results for the original Hamiltonian and for the transformed Hamiltonian obtained using wave function matching. We perform the benchmark for radii $R = 2.0$ fm, $R = 4.6$ fm, and $R = 7.2$ fm. We confirm that the low-energy physics of wave function matching is independent of R . We also confirm again that the induced three-body interaction is numerically small.

While the radius R does not change the low-energy physics, it does change the high-energy physics near the momentum cutoff scale. In Methods, we have written yet another new section called “New Chiral Effective Field Theory Interactions with Wave Function Matching”. There we write:

“We have discussed many computational aspects of wave function matching and its ability to accelerate the convergence of perturbation theory in *ab initio* many-body calculations. In this section, we discuss the use of wave function matching as a theoretical tool for exploring new chiral effective field theory interactions with different short-distance regulator properties. By changing the high-fidelity Hamiltonian H , the simple Hamiltonian H^S , the wave function matching radius R , and/or the choice of unitary transformation U (Gram-Schmidt orthogonalization, Givens rotation, etc.), we can construct a large class of transformed Hamiltonians H' . As an important bonus, the accelerated convergence of perturbation theory provides an efficient method for performing the calculations. This allows for the exploration of a very large class of high-fidelity chiral interactions with a wide range of different short-distance regulators.

In Fig. S5, we showed that the ${}^3\text{H}$ and ${}^4\text{He}$ binding energies changed very little when varying R from 3.22 fm to 3.97 fm. We argued in Section S8 that when R exceeds r_Δ , the dependence on R only appears in high-momentum modes. While there is no noticeable effect on the low-energy physics of few-nucleon systems, these high-momentum modes can impact nuclear many-body properties. As we have argued in Section S13, this may happen if the $\alpha\alpha$ interaction and/or other cluster interactions are impacted by the change in R . This is in fact what happens.

As R is varied from 3.22 fm to 3.97 fm, the binding energy of ${}^{12}\text{C}$ decreases by 6 MeV while the binding energy ${}^{16}\text{O}$ decreases by 12 MeV. This is the physics of the quantum phase transition studied in Ref.⁷⁰ By changing the $\alpha\alpha$ interaction, one can produce a quantum phase transition from a nuclear liquid to a Bose gas of α particles. In that study, the two-nucleon phase shifts and few-nucleon properties were kept approximately the same. Here, the two-nucleon phase shifts remain exactly the same, while the binding energies for ${}^3\text{H}$ and ${}^4\text{He}$ change by no more than 1%. Wave function matching provides a new theoretical tool for exploring new chiral interactions with different short-distance regulators. Similar to what we have done in this work, we expect that the dependence on R in nuclear many-body systems can be removed by tuning the regulator structure of the three-nucleon interactions. This work is currently in progress. It is part of a larger program to show that one can produce different high-fidelity chiral interactions with different regulator structures using wave function matching and then cancel regulator-dependent errors by tuning the regulator structure of the three-nucleon interactions.”

3. The “short-range” three-body contact from equation S21 couples nucleons that are two lattice spacings apart. This does not seem to be a short-range force. This term is quite important for saturation, and deviates from chiral EFT.

This is a good question. In Methods, we now write:

“Although the three nucleons sit on different lattice sites, the three-nucleon configurations for $V_{CE}^{(l)}$ and $V_{CE}^{(t)}$ are still very compact. The root-mean-square radius for the $V_{CE}^{(l)}$ configuration is 1.07 fm, while the root-mean-square radius for the $V_{CE}^{(t)}$ configuration is 0.76 fm.”

If nucleons were hard spheres with radius 0.66 fm, then the configuration where three nucleons are touching and arranged along a line would correspond to the configuration for $V_{CE}^{(l)}$. A two-nucleon interaction at distance 2.64 fm that couples to low-energy nucleon modes would scramble the low-energy convergence of effective field theory. However, we are requiring that a third nucleon sits in between the

two outer nucleons, and there is empirical evidence that this is a compact configuration. For calculations of ${}^3\text{H}$ and ${}^4\text{He}$, the ratios of contributions for the various three-nucleon interactions are

$$\frac{\langle V_{cD}^{(0)} \rangle_{4\text{He}}}{\langle V_{cD}^{(0)} \rangle_{3\text{H}}} = 4.35(1), \quad \frac{\langle V_{cD}^{(1)} \rangle_{4\text{He}}}{\langle V_{cD}^{(1)} \rangle_{3\text{H}}} = 4.38(1), \quad \frac{\langle V_{cD}^{(2)} \rangle_{4\text{He}}}{\langle V_{cD}^{(2)} \rangle_{3\text{H}}} = 4.40(1),$$

$$\frac{\langle V_{cE}^{(0)} \rangle_{4\text{He}}}{\langle V_{cE}^{(0)} \rangle_{3\text{H}}} = 4.29(1), \quad \frac{\langle V_{cE}^{(1)} \rangle_{4\text{He}}}{\langle V_{cE}^{(1)} \rangle_{3\text{H}}} = 4.33(1), \quad \frac{\langle V_{cE}^{(2)} \rangle_{4\text{He}}}{\langle V_{cE}^{(2)} \rangle_{3\text{H}}} = 4.36(1), \quad \frac{\langle V_{cE}^{(l)} \rangle_{4\text{He}}}{\langle V_{cE}^{(l)} \rangle_{3\text{H}}} = 4.41(1), \quad \frac{\langle V_{cE}^{(t)} \rangle_{4\text{He}}}{\langle V_{cE}^{(t)} \rangle_{3\text{H}}} = 4.55(1).$$

We see that all of the ratios agree within a few percent and are consistent with the universal Tjon band physics associated with short-ranged three-nucleon interactions. Three-nucleon interactions with longer range produce different results that deviate from this universal behavior. For example, the longer-ranged two-pion exchange three-nucleon interaction $V_{3\text{N}}^{(\text{TPE2})}$ produces a somewhat larger ratio of 4.63(2).

The link between nuclear saturation and the locally-smearred three-nucleon interactions is not specific to the $V_{cE}^{(l)}$ interaction. In Methods, we now write:

“The strong correlation between accurate nuclear binding energies and the interactions among α particles and nucleons can be seen empirically from the data in Tables S4-S10. In Table S4, we see that using only the simplest three-body interactions $V_{cE}^{(0)}$ and $V_{cD}^{(0)}$ does a poor job in reproducing the nuclear binding energies. The fundamental problem can be diagnosed quite easily as arising from the fact that energy differences such as $E_{12,0_1^+}^6 - 3E_4^2$, $E_{16}^8 - 4E_4^2$, $E_6^2 - E_4^2$, $E_9^4 - E_8^4$, $E_{13}^6 - E_{12}^6$, and $E_{14}^6 - E_{12}^6$ are significantly higher than they should be. Given the known cluster structure for many of these light nuclei, we conclude that the interactions among the α particles and between α particles and nucleons need to be more attractive.”

Later we write:

“Given the importance of the $\alpha\alpha$ interaction for the binding of nuclei with equal numbers of protons and neutrons, we expect that the first parameter that must be tuned is the $\alpha\alpha$ interaction. This statement was already demonstrated in Ref.⁵⁰ In that work, three of the interaction parameters were tuned according to the properties of few-body systems with up to three nucleons. By tuning only one additional parameter associated with the strength of the local smearing regulator, the properties of light nuclei, medium-mass nuclei, and neutron matter were reproduced with no more than a few percent error. The success of this simple approach provides evidence that one parameter, the strength of the $\alpha\alpha$ interaction, plays a dominant role in the binding of nuclei with equal numbers of protons and neutrons.

We can see the same underlying physics in our analysis here. Comparing Table S4 and Table S5, we see that adding just one additional three-nucleon interaction with local smearing increases the binding of nuclei with even and equal numbers of protons and neutrons. With the exception of the last row in Table S5 corresponding to the poorest fit, we see that the fits using $V_{cD}^{(2)}$, $V_{cE}^{(1)}$, $V_{cE}^{(2)}$, $V_{cE}^{(l)}$, and $V_{cE}^{(t)}$ all have similar features. This can be seen in the lowering of the energy differences $E_{12,0_1^+}^6 - 3E_4^2$ and $E_{16}^8 - 4E_4^2$, which are strongly correlated with the $\alpha\alpha$ interaction. In each case, the energy differences $E_{12,0_1^+}^6 - 3E_4^2$

and $E_{16}^8 - 4E_4^2$ are brought much closer to their physical values, and the root-mean-square error for the binding energy per nucleon drops from 1.2 MeV to about 0.4 MeV. ”

In order to get proper nuclear saturation, the $\alpha\alpha$ interaction must be close to the physically-observed strength. The $V_{CE}^{(l)}$ interaction has no special role. Any locally smeared interaction can be used to tune the $\alpha\alpha$ interaction.

4. Are the values for the six additional three-nucleon low-energy constants natural in size?

We thank the referee for this question. The three-nucleon low-energy constants are natural in size. In Methods, we now write:

“In the right panel of Fig. 3, we see that the energy per nucleon with two-nucleon interactions only does not go below -10 MeV. This can be interpreted as the $\alpha\alpha$ interaction not having enough attraction. Similar to what we found in Table S5, the addition of almost any locally smeared three-nucleon interaction fixes the problem quite easily. The size of this correction is about 6 MeV per nucleon at the saturation density. Given that the three-nucleon interactions appear at NNLO or $O(Q^3)$ and the Fermi momentum of about 260 MeV, this contribution is of natural size if the expansion parameter is Q/Λ with $\Lambda \sim \pi/a = 471$ MeV.

For light nuclei, the relevant momentum scale Q is approximately the pion mass. The natural size for the contribution of the three-nucleon interactions is about 1 MeV per nucleon. This is indeed what is observed in the lattice calculations. For medium-mass nuclei, the three-nucleon contribution to the binding energy is intermediate between the 1 MeV per nucleon for light nuclei and 6 MeV per nucleon at the saturation density.”

Depending on the specific fit, one energy correction may be bigger than another, especially if one specific interaction is used to adjust the $\alpha\alpha$ interaction. But none of the three-nucleon interactions are unusually large or unusually small in comparison with the other three-nucleon interactions. All three-nucleon interactions are consistent with having natural size.

5. The approach seems to have little predictive power. 17 nuclei, including calcium-40, were included in the fit of the three-nucleon force.

In the General Comments at the beginning of this document, we list around 50 new predictions that stem from this work without fitting any additional parameters. There is no limit to the number of other predictions for nuclear structure, reactions, scattering, and thermodynamics that follow immediately from this work. Many of these projects are already planned for the next few years. Predictive power is not a problem, and the prediction quality appears to be very good.

We have used 17 nuclei to fit six parameters characterizing the local smearing regulators for the three-nucleon force. From this analysis, we have learned that with three or four parameters, the error can be reduced to the level of 0.1 MeV per nucleon. With this new information in hand, one can now restart with fewer parameters and fit the binding energies of fewer nuclei. This procedure has been performed and the results are now included in Methods. They are presented in Table S19 and also listed in the response to Referee 3.

6. *It is not clear from Fig. 3 that nuclear matter saturates at an accurate point.*

We thank the referee for this constructive comment. In the main text we now write:

“Within the uncertainties due to finite system size corrections, the symmetric nuclear matter calculations show saturation at an energy and density consistent with the empirical saturation point labeled with the black rectangular box. The relative uncertainties due to finite system size are at the 10% level for the energy. Additional calculations with larger systems are needed to reduce the thermodynamic extrapolation error further.”

D. Appropriate use of statistics and treatment of uncertainties

The authors use state-of-the-art tools such as emulators and history matching to arrive at their results. They only give statistical uncertainties on their main results. One wonders what the sensitivities are with respect to the three-body parameters and the matching distance R ?

We thank the referee for these questions. The list of error sources included in the reported errors for the binding energies was incomplete. The uncertainty associated with the interactions were part of these error estimates. In the revised main text we now write:

“The nuclei used in the fit of the three-nucleon interactions are labeled with open squares, while the other nuclei are predictions denoted with filled diamonds. The one-standard-deviation posterior distribution error bars shown in Fig. 2 represent uncertainties due to Monte Carlo errors, infinite-volume extrapolations, infinite projection time extrapolations, and uncertainties associated with the interactions.”

The sensitivity to the wave function matching radius R is now addressed and is detailed above in the reply to point 2.

E. Conclusions: robustness, validity, reliability

The abstract contains incorrect or misleading statements (see point A above). The results certainly seem valid, but they are not in the framework of chiral EFT.

These issues are now fixed as detailed above in the reply to point A.

F. Suggested improvements: experiments, data for possible revision

1. *Addressing the questions raised under points C and D.* 2. *Include the formula used to compute the*

charge radii. Did the authors include the spin-orbit correction?

The points C and D are addressed above. We thank the referee for asking for our formula for calculating the charge radii. In Methods we now write:

“In our calculations of the root-mean-square charge radii, we use the relation,

$$\langle r_{\text{ch}}^2 \rangle = \langle r_{pp}^2 \rangle + R_p^2 + \frac{N}{Z} R_n^2 + \frac{3}{4m_p^2}, \quad (1)$$

where $\langle r_{pp}^2 \rangle$ is the mean-square point proton radius, R_p^2 is 0.7056 fm^2 ,^{133,134} R_n^2 is -0.105 fm^2 ,¹³⁵ and m_p is 938.27 MeV . We have not included relativistic spin-orbit corrections or two-nucleon current contributions. As the size of the computational errors for the lattice calculations are reduced in the future, these smaller contributions will also be included.”

G. References: appropriate credit to previous work?

Missing references are listed under point B.

We thank the referee for suggesting these excellent references. They are now included in the bibliography.

H. Clarity and context: lucidity of abstract/summary, appropriateness of abstract, introduction and conclusions

The paper is well written, with the caveats presented in points B and D.

The caveats in points B and D have been addressed above. We thank the referee for the positive comments and the many useful suggestions and questions that have pushed us to improve the impact and lasting importance of this work.

Reply to Reviewer 2

In the manuscript « Wave function matching for the quantum many-body problem », the authors, Elhatisari et al., present a new method to compute the structure of many-body quantal systems. In particular, they look at light- to medium-mass nuclei. These structures are difficult to compute because of the complexity of the interaction between the nucleons. Finding the eigenstates of modern, realistic nuclear Hamiltonians is therefore quite a difficult task. The idea behind the « wave-function matching » method is to transform the realistic Hamiltonian H into another Hamiltonian H' through a unitary transformation U . That new Hamiltonian therefore exhibits the exact same physics content as the initial H , but it is chosen so that the short-range behaviour of its wave functions is proportional to that of a simple many-body Hamiltonian H^S of which the eigenstates can be easily computed. The particular unitary transformation U that enables such properties of the eigenstate wave functions of H' is the Hamiltonian translator (S1) applied between H^S and H .

The present manuscript shows the results obtained with this new, innovative, method on light- to medium-mass nuclei computed with Monte-Carlo Quantum calculations. Excellent agreement is obtained with experiment, even on nuclei not used to fit the low-energy constants of the nuclear interaction considered by the authors. This suggests the method has a good predictive power. Unfortunately, the manuscript lacks clarity and it is not always clear what has been done. I understand that within a short text, it is not possible to include all the necessary details to be able to reproduce the authors' calculations. Nevertheless, that is the purpose of the « Methods » section.

We fully agree with these comments. We have significantly improved the clarity of the main text and greatly augmented the Methods section with details of calculations, discussions of concepts, and several different benchmark examples.

The authors should address the following issues prior to a possible acceptance of their manuscript.

My major concern is on the test of the method. Would it be possible to confront this new method to other many-body techniques using one given nuclear interaction? Such a test could be performed on a small system like 4He , which is accessible to various exact methods. The authors seem to make this test, but only on the deuteron, of which the structure is not affected by three-body forces, and the test illustrated in the right panel of Fig. 1 on the triton and 4He does not include three-body forces. Accordingly, the N3LO calculations (solid symbols) does not reproduce the data (black open box). To demonstrate that this new method works, a fully converged calculation using the wave function matching should be compared to another « exact » method using exactly the same nuclear interaction. I guess that a comparison with the no-core shell model or the Green's function Monte Carlo method could be done using the N3LO chiral EFT interaction considered by the authors. Can this test be performed? In the affirmative, it should be shown to prove the validity of the approach. If not, it would be difficult to accept this as a reliable way to compute many-body quantal systems.

We thank the referee for this excellent suggestion. The referee would like to see a rigorous benchmark for a system with at least three particles where a fully-converged calculation with wave function matching is compared against a fully-converged calculation without wave function matching. Such comparisons are often done with similarity renormalization group calculations, and one typically sees that the inclusion of the induced three-nucleon interactions give very good agreement between the two calculations.

For a high-accuracy benchmark test, however, we need the starting Hamiltonians to be identical for the two calculations. This makes the comparison of a lattice calculation against a continuous space calculation a poor choice. They do not use the same starting Hamiltonians. It is much better to compare two calculations on the lattice or two calculations in continuous space.

In Methods, we have written a new section called “Induced Three-Body Interactions in Wave Function Matching: Born-Oppenheimer Analysis”. We benchmark a three-body system on a three-dimensional lattice where two of the particles have infinite mass and the third particle has finite mass. For all possible separation distances between the two infinite-mass particles, we compare the ground state energies for the original Hamiltonian H and the transformed Hamiltonian H' obtained using wave function matching. The difference between the three-body energies gives the induced three-body interaction as a function of distance between the infinite-mass particles. The results are computed for different values of the wave function matching radius, R . We show results for $R = 1.5$ lattice units in Fig. S6, $R = 2.5$ lattice units in

Fig. S7, $R = 3.5$ lattice units in Fig. S8, $R = 4.5$ lattice units in Fig. S9, and $R = 5.5$ lattice units in Fig. S10.

Several things can be learned from these results. The first is that the induced three-body interactions are numerically very small. The second is that the induced three-body interactions have the same short range for all values of R . Both of these features are very different from the induced three-body interactions seen for the similarity renormalization group. For renormalization group evolution, the induced three-body interactions become larger and more long ranged as the resolution scale is lowered. The theoretical basis for the short range of the induced three-body interaction and the independence of R is explained in section S8:

“The reason that the induced three-body interaction remains small in magnitude comes from the fact that H , H^S , H' all have non-singular interactions with a short range. We can use naive dimensional analysis to predict that the relative strength of the induced three-body interaction in d spatial dimensions is suppressed by $(pr)^d$, where p is the typical particle momentum and r is the interaction range. This example shows that the wave function matching calculation is under good control. The induced three-body interaction is numerically small and has a range that is short and independent of R .”

This benchmark example provides numerical confirmation that, when applying wave function matching to chiral effective field theory, the transformed Hamiltonian H' can be viewed as a new chiral effective field Hamiltonian. There are no long-range interactions or large induced three-nucleon interactions. Instead, the induced three-nucleon interactions are small and short ranged. In fact, wave function matching provides a useful theoretical tool for generating new chiral effective field Hamiltonians.

In Methods, we have also written another new section called “Induced Three-Body Interactions in Wave Function Matching: Three Particles in One Dimension”. There we study a three-body system composed of three distinguishable particles on a one-dimensional lattice. We compute the six lowest three-body energies in a periodic box and compare results for the original Hamiltonian and for the transformed Hamiltonian obtained using wave function matching. We perform the benchmark for radii $R = 2.0$ fm, $R = 4.6$ fm, and $R = 7.2$ fm. We confirm that the low-energy physics of wave function matching is independent of R . This example again shows that the wave function matching calculation is under good control. The induced three-body interaction is numerically small and is independent of R . In this example, we also demonstrate that the convergence of perturbation theory for the three-particle system is greatly accelerated by wave function matching.

Wave function matching is proving its value as a powerful and reliable many-body method for nuclear lattice effective field theory. This is demonstrated by the examples presented in this work and the 50 additional examples presented above in the General Comments. For the greatest possible impact, however, it should be extended to continuous space so that it becomes accessible to the broad scientific community. There are immediate applications for quantum Monte Carlo simulations in continuous space for nuclear physics, quantum chemistry, ultracold atoms, and condensed matter. It should also have important applications to other *ab initio* calculations as well as quantum computing algorithms. In Methods, we have written a new section called “Wave Function Matching in Continuous Space”. We show how to implement wave function matching efficiently in continuous space using spatially-compact orthogonal basis states such as the eigenstates of a harmonic oscillator.

*Related to the previous issue, as in other nuclear-structure techniques, the wave function matching induces many-body forces (see the one-before-last paragraph on p. 4). This leads the authors to determine new coefficients for, e.g., three-body forces, see p. 11 in the « Methods » section. If these parameters have to be re-fitted, what is the actual predictive power of the wave function matching? Do I understand correctly that a calculation using, e.g., a given chiral EFT nuclear interaction with three-body forces, would not lead to accurate binding energies for $A > 2$ nuclei? Accordingly, could these calculations really be called « *ab initio* »?*

We thank the referee for this question. We have already discussed the new section “Induced Three-Body Interactions in Wave Function Matching: Born-Oppenheimer Analysis”, where we show that the induced three-body interaction is numerically small and has a range that is short and independent of R . In the new section “Induced Three-Body Interactions in Wave Function Matching: Three Particles in One Dimension”, we again show that the induced three-body interaction is numerically small and is independent of R . This shows that wave function matching produces a new high-fidelity chiral Hamiltonian H' that is just as good as H . The phase shifts for H' are excellent, the same as for H . The range of the interaction in H' for low-energy physics is the same as that for H and H^S , and the Tjon universality for ${}^3\text{H}$ and ${}^4\text{He}$ is satisfied. There is no need to “refit” anything, and there is no need to compute induced three-body forces. We just use H' as our starting two-nucleon Hamiltonian and proceed as we would with any other chiral Hamiltonian.

In the new manuscript we add some clarifying remarks stating that wave function matching has no direct connection to the strategy for reducing errors in *ab initio* nuclear structure calculations:

“Before proceeding to larger nuclei and many-body systems, we first comment on the current status of *ab initio* calculations of nuclear structure using χEFT . The following analysis is not directly connected to wave function matching. Instead, it is a separate theoretical framework designed to help push beyond the current limitations of *ab initio* nuclear structure theory.”

We could simply use wave function matching to produce a chiral Hamiltonian H' and use the standard *ab initio* approach of choosing a three-nucleon regulator in some arbitrary fashion and fitting c_D and c_E to few-body observables. If the regulator has some amount of local smearing, but not too much, this will work quite well, probably as well as the best *ab initio* calculations currently available. The reason we don’t take this path is because we are trying to diagnose and fix the problems currently thwarting progress in *ab initio* nuclear structure theory. We are trying to push beyond the current limitations. In the new manuscript we write:

“Current *ab initio* calculations have difficulty simultaneously maintaining high-fidelity two-nucleon phase shifts and describing the saturation energy and density of symmetric nuclear matter as well as the binding energies and charge radii of light and medium-mass nuclei. Previous *ab initio* nuclear structure calculations have either not addressed some of the relevant observables or require further improvement in one or more of these areas. We aim to identify the problem and point to a viable solution.”

The implementation of the method is also not very clear. The authors mention an « expansion in powers of the difference $H' - H^S$ » (see the bottom of p. 1). Does this mean that the translator (SI) is

expanded in powers of H_T , which is a difference of two Hamiltonians? If yes, wouldn't a Padé expansion be more appropriate to conserve the unitarity of the translator? The calculations shown in the right panel of Fig. 1 are performed at the first order of that expansion. What about the results shown in Figs. 2 and 3? Are they also obtained at the sole first order of that expansion or were they obtained with higher orders of the expansion scheme?

We thank the referee for this question. In Methods, the updated section “Wave Function Matching” explicitly constructs the unitary transformation for three cases: Gram-Schmidt orthogonalization, Householder reflection, and Givens rotation. These examples should make it more clear that we are not using perturbation to define the two-body unitary transformation. The transformations are exact two-body unitary transformations. Perturbation theory is only used when computing eigenstates and energies of H' starting from the eigenstates of H^S .

The results shown in the right panel of Fig. 1 are computed using first-order perturbation theory. The fact that the wave function matching results satisfy Tjon universality is a sign that perturbation theory is converging very rapidly. The results computed without wave function matching violates Tjon universality. This is because first-order perturbation theory for that case is not reliable.

The results shown in Fig. 2 and Fig. 3 also use first-order perturbation theory. In order to reduce errors further, we allow the couplings of the Hamiltonian used to prepare the many-body wave function to be free parameters, and use the variational principle to get the best possible approximation to the many-body wave function. To clarify, in Methods we now write:

“For the many-body calculations, we perform the following steps. We first write the transformed two-nucleon Hamiltonian H' as $H' = H^S + (H' - H^S)$. We then prepare eigenstates of H^S and apply corrections up to first order in perturbation theory to get the properties of the eigenstates of H' . The three-nucleon interactions are added to H' at this stage. In order to accelerate the convergence of perturbation theory further, we consider the more general partition $H' = H'^S + (H' - H'^S)$. The modified simple Hamiltonian H'^S has the same form as H^S in Eq. (S1), but we allow for different coupling strengths $c_{\text{SU}(4)}$ and c_I . We then minimize the energy and use the variational principle to optimize the parameters.”

Some examples confirming the high quality of these variational calculations are shown in Table S18.

On p. 3, the authors make a comment on cluster EFT (see the third paragraph). I do not really see how this discussion on the cluster EFT fits within this work. Were the calculations obtained assuming an alpha-cluster structure of the nuclei, similar to the Ikeda diagram (plus maybe a few nucleons when N does not equal Z)? [see Ikeda et al. Prog. Theor. Phys. Suppl. 464 (1968)] Or are the calculations shown here full A -body calculations, where A is the mass number?

We thank the referee for pointing out the lack of clarity in this part of the manuscript. All of the calculations in this work are full A -body calculations. Cluster effective field theory is only used to predict the maximum number of parameters we will need to tune in order to cancel regulator-dependent errors from the *ab initio* nuclear many-body calculations. In the main text, we now write:

“Using the formalism of cluster effective field theory^{30–33} for α particles and nucleons, the two-cluster

interactions are $\alpha\alpha$ and αN , and the three-cluster interactions are $\alpha\alpha\alpha$, $\alpha\alpha N$, and αNN , with αNN having two possible isospin channels. In addition to the $\alpha\alpha$ interaction, there may be some dependence on short-distance physics arising from these other cluster interactions. This gives us a simple counting argument that there is one important parameter associated with the $\alpha\alpha$ interaction and at most five other parameters that also require tuning to reduce unwanted errors. Our strategy is to tune the short-distance features of the three-nucleon interactions to achieve this error cancelation.”

In the subsection « Wave Function Matching » of the « Methods » section, the authors « restrict [themselves] on the space of two nucleons ». What is done in actual calculations? Is the wave-function-matching method applied only to the two-body part of the nuclear interaction, or does it affect also the many-body wave function?

We thank the referee for asking this question. It points to a lack of clarity in the original manuscript. Wave function matching is used to define the new two-nucleon Hamiltonian H' . In Methods we now make the explanation of the steps for the many-body calculation more explicit:

“For the many-body calculations, we perform the following steps. We first write the transformed two-nucleon Hamiltonian H' as $H' = H^S + (H' - H^S)$. We then prepare eigenstates of H^S and apply corrections up to first order in perturbation theory to get the properties of the eigenstates of H' . The three-nucleon interactions are added to H' at this stage. In order to accelerate the convergence of perturbation theory further, we consider the more general partition $H' = H'^S + (H' - H'^S)$. The modified simple Hamiltonian H'^S has the same form as H^S in Eq. (S1), but we allow for different coupling strengths $c_{\text{SU}(4)}$ and c_I . We then minimize the energy to optimize the parameters. We should clarify that the parameters $c_{\text{SU}(4)}$ and c_I in H'^S are only used to improve the quality of the variational trial states and have nothing to do with the actual interaction parameters of the Hamiltonian.”

In addition to these main questions, I have other comments on the manuscript; there remain some typos and some sentences could be improved to ease the reading of the text:

We thank the referee for catching each of these errors.

- p. 2, end of the paragraph below Fig. 1: « the separation distance beyond which » (replace « at » by « beyond »?)

This correction has now been made.

- p. 2, last line: « (NLO). » (remove one of the two full stops).

This correction has now been made.

- p. 3, middle of the page: « At order N3LO, there are additional terms ... at N3LO order »; one of the « N3LO » should be removed. Note also that O in N3LO means « order », so « N3LO order » is redundant. Same comment in the caption of Fig. 2.

This correction has now been made.

- *To what does the black open box correspond in the right panel of Fig. 3?*

We thank the referee for this question. The caption in the right panel of Fig. 3 now reads:

“The empirical saturation point is labeled with a black rectangular box.”

In the main text, we also write:

“Within the uncertainties due to finite system size corrections, the symmetric nuclear matter calculations show saturation at an energy and density consistent with the empirical saturation point labeled with the black rectangular box.”

- *p. 7, 3rd paragraph, after its definition, the function $\langle f(t) \rangle$ has an argument t/T , which is not consistent. Actually, $\langle f(t) \rangle$ is the value of the function in t , so writing $\langle \text{Let } f \text{ be a smooth function.} \dots \rangle$ would be enough.*

This is an excellent suggestion. The change has now been made.

- *In the subsection $\langle \text{Wave Function Matching} \rangle$, it would help to relate H , H^S and H' to the Hamiltonians H_A , H_B , H_A' and H_B' of the previous subsection $\langle \text{Hamiltonian Translators} \rangle$. This would help the reader understand what is actually done in this new method. If the authors develop (S1) in powers of H_T , this would also be good place to mention it.*

We thank the referee for this suggestion. Since H , H^S and H' are used in the main text and throughout the article, we have added the following text to the section “Hamiltonian Translators”:

“Before describing how wave function matching is implemented in practice, we first discuss a class of transformations called Hamiltonian translators. Let H_A and H_B be two Hamiltonians acting on the same linear space. H_B corresponds to the simple Hamiltonian that we called H^S in the main text and H_A corresponds to the high-fidelity Hamiltonian H . We have temporarily changed the notation here so that we can use a convenient notation with A and B as subscripts.”

- *p. 9, first line below Eq. (S17): K has already been defined on the previous page [see below Eq. (S4)].*

This correction has now been made.

- *p. 10, the authors mention for the first time the $\langle \text{history matching} \rangle$. Unfortunately, this method is defined only at the bottom of p. 11, and the corresponding Ref. 60 is provided only on p. 13. This should be revised.*

We thank the referee for this suggestion. The first mention of history matching now includes a description of what it is and all of the history matching references:

“As we discuss in the following, we treat these LECs as unknown regression coefficients of an emulator employed with history matching, which has been shown to be an effective approach for parameter searches.^{22,59–61} History matching is an iterative process that identifies and eliminates implausible parts of the input space by measuring implausibility of inputs, shrinks the input space in every iteration, and repeats the non-implausible input search in the smaller input space.

- p. 10, two lines above Eq. (S26): $\ll f(x) \gg$ is mentioned but Eq. (S26) define f_o . Is this a typo?

The corrected material now reads:

“For each interaction, we approximately collect 10^3 samples. Then we can use $2N$ LECs from these distributions as inputs, which form a list represented by a vector \mathbf{x} where the individual inputs are x_i with $i = 1, \dots, d$. We define our *ab-initio* model for any number of outputs and signify by $f_o(\mathbf{x}, \beta)$ with the individual outputs $o = 1, 2, \dots, q$ and adjustable parameters in the vector β . Now we define the following emulator equation for our *ab initio* model,

$$f_o(\mathbf{x}, \beta) = \sum_j \mathbf{x}_j \left(\frac{\partial E}{\partial \mathbf{x}} \right)_{jo} + \sum_k \beta_k \left(\frac{\partial E}{\partial \beta} \right)_{ko}. \quad (2)$$

The first term on the right hand side is the model output of only $2N$ interactions. This term uses $2N$ inputs from the distributions shown in Fig. S3, which are denoted by the vector \mathbf{x}_j where the subscript j denotes different $2N$ interactions.”

- p. 11, first paragraph: to what do $\ll V_{cE}^{(0)}, V_{cE}^{(1)}, V_{cE}^{(2)}, V_{cD}^{(0)}, V_{cD}^{(1)}, \text{ and } V_{cD}^{(2)} \gg$ correspond? Are they related to the expressions detailed in Eqs. (S20), (S21), and (S22), or do they correspond to other three-body forces? In the latter case, wouldn't it be useful to include the expression and physical meaning of these forces? Note that a short physical explanation of the difference between $V^{(l)}_{cE}$, $V^{(t)}_{cE}$ etc. would be helpful.

We thank the referee for these suggestions. We now make the link to the relevant equations appearing earlier:

“For this term we consider the SU(4) symmetric three-nucleon forces $V_{cE}^{(0)}, V_{cE}^{(1)}, V_{cE}^{(2)}, V_{cE}^{(l)}, V_{cE}^{(t)}$ given in Eqs. (S16), (S18), (S19), and the one-pion-exchange three-nucleon forces $V_{cD}^{(0)}, V_{cD}^{(1)}$ and $V_{cD}^{(2)}$ defined in Eq. (S17).”

We now also give the following descriptions:

“In Fig. S14, $V_{cD}^{(0)}$ shown in panel a is the c_D term with no additional local smearing. $V_{cD}^{(1)}$ in panel b is the c_D term with local smearing up to one lattice spacing. $V_{cD}^{(2)}$ in panel c is the c_D term with local smearing up to $\sqrt{2}$ times the lattice spacing. $V_{cE}^{(0)}$ in panel d is the c_E term with no additional local

smearing. $V_{c_E}^{(1)}$ in panel e is the c_E term with local smearing up to one lattice spacing. $V_{c_E}^{(2)}$ in panel f is the c_E term with local smearing up to $\sqrt{2}$ times the lattice spacing. $V_{c_E}^{(l)}$ in panel g is the c_E term with local smearing where the three nucleons are at different sites with a prolate configuration. This corresponds to three nucleons lying along a line with each nucleon one lattice unit apart from their neighbor(s). $V_{c_E}^{(t)}$ in panel h is the c_E term with local smearing where the three nucleons are on different sites with an oblate configuration. This corresponds to three nucleons forming an equilateral triangle with side length equal to $\sqrt{2}$ lattice units.”

- p. 14, within the paragraph « WAVE 3 », there are weird spaces after « WAVE 2 » and « WAVE 3 » within the text.

This correction has now been made.

- p. 14, third line of the paragraph « WAVE 3 »: « we run the implausibility measure » (one « the » too many).

This correction has now been made.

- p. 16, third line: « clearly indicate » should be at the plural form because the subject is « The first few rows ».

This correction has now been made.

In conclusion, this manuscript presents the wave function matching, a new method to compute accurately many-body quantal systems. Its application is illustrated here on the complex and difficult problem of atomic nuclei. However, if proved exact, it could probably be very useful beyond the domain of nuclear physics, for example in atomic or solid-state physics. Unfortunately, and probably because of the short length of the manuscript, the details on the method and its actual implementation are not very clear. The authors should address the different issues raised in this report to make their manuscript accessible to the broad readership of Nature.

We thank the referee for the encouraging comments on the potential impact of wave function matching to nuclear physics and other fields. We appreciate the many helpful questions and suggestions. We have improved the clarity of concepts and readability of the main text, with an eye towards making the manuscript accessible and interesting to the broad readership of Nature. We have also added a considerable amount of new material to Methods, providing details of calculations and discussions of concepts. We have also presented several benchmark examples that provide numerical evidence that wave function matching is performing as advertised.

Reply to Reviewer 3

The Authors present a new method called wave function matching that is designed to quench sign problems in lattice Monte Carlo calculations. They use this method with high-fidelity chiral EFT interactions and lattice EFT to compute energies and radii in finite nuclei and the equation of state for infinite nuclear & neutron matter. The Authors claim to provide some insights that may help resolve long-standing challenges in accurately reproducing bulk properties of nuclei and saturation of nuclear matter.

On the one hand, the method of wave function matching, accompanied by six additional and non-standard short-range terms in chiral EFT, is original and very creative. The fact that this approach solves the problem of sign cancellations of high-fidelity Hamiltonians on the lattice and predict good radii, once the short-range terms are fitted to energies in heavy systems, is certainly an accomplishment. The manuscript presents technical developments that I believe is of interest to the broader physics community as well as developers & practitioners of Monte Carlo methods.

On the other hand, I am unsure whether this technical development warrants publication in Nature and I have a series of concerns that cast doubt on the physics conclusions that can be drawn from the presented results. In particular regarding the predictive power, physics insight, and systematic improvement once the additional short-range terms have been introduced. Do these terms solve a technical Monte Carlo problem or a physics problem? Can the Authors present evidence that clarifies this question and strengthens the physics conclusions in the manuscript?

We thank the referee for the excellent questions. Wave function matching is a new theoretical paradigm for quantum many-body systems. From the examples presented in this work and the additional examples in the General Comments, it is already fulfilling its promise as a powerful computational tool for *ab initio* nuclear lattice effective field theory. There are many potential applications for continuous-space quantum Monte Carlo simulations in nuclear physics, quantum chemistry, ultracold atoms, and condensed matter. For this purpose, we have written a new section in Methods called “Wave Function Matching in Continuous Space”.

While the many computational applications may grab more popular attention, wave function matching is more than a technical development. It is a brand new way to think about the quantum many-body problem, and some of the original ideas are explored in the revised manuscript. In the section “Hamiltonian Translators”, we present the theoretical foundations of wave function matching and introduce the concept of a Hamiltonian translator. Suppose that U_{AB} is a unitary transformation mapping all the eigenvectors of H_B to all the eigenvectors of H_A . We then call U_{AB} a Hamiltonian translator from H_B to H_A . The transformed Hamiltonian $H'_A = U_{AB}^\dagger H_A U_{AB}$ is a Hamiltonian with energy eigenvalues identical to those of H_A , but with eigenvectors identical to those of H_B . If we write $H'_A = H_B + (H'_A - H_B)$, then the zeroth-order expansion of the eigenvectors is exact, and the first-order expansion of the energies is exact. Wave function matching is an approximate Hamiltonian translator at the two-body level.

We can also make a connection with adiabatic quantum computing. If we have a time-dependent Hamiltonian $H(t)$ that slowly evolves from H_B at time $t = 0$ to H_A at time $t = T$, then the time evolution operator from $t = 0$ and $t = T$ is an exact Hamiltonian translator U_{AB} in the limit that T goes to infinity. The connection between quantum computing and wave function matching doesn't stop there. In the conclusions of the main text, we now write:

“While we have focused on Monte Carlo simulations for nuclear physics here, wave function matching can be used with any computational method and applied to any quantum many-body system. This also includes quantum computing algorithms where wave function matching can be used to reduce the number of quantum gates required. All that is needed is a simple Hamiltonian H^S that produces fair agreement with empirical data for the many-body system of interest and is easily computable using the method of choice.”

In Methods we have added a new section called “Analyticity of Wave Function Matching”, which shows that wave function matching does not introduce any non-analytic behavior in momentum space. We also explore in great detail the question of how wave function matching depends on the radius R . We derive the fascinating and surprising result that the low-energy physics is independent of R . This is new physics that has not been explored previously and shows that wave function matching is a new theoretical construction that has nothing to do with renormalization group evolution. It is doing something different. The theory of wave function matching may become a topic of broad scientific interest.

In Methods, we have also written a new section called, “New Chiral Effective Field Theory Interactions with Wave Function Matching.” In that section we discuss the use of wave function matching as a theoretical tool for exploring new chiral effective field theory interactions with different short-distance regulator properties. While we use nuclear theory as the prototype example in this section, the conclusions apply to any low-energy effective field theory and its application to quantum many-body systems. We write:

“By changing the high-fidelity Hamiltonian H , the simple Hamiltonian H^S , the wave function matching radius R , and/or the choice of unitary transformation U (Gram-Schmidt orthogonalization, Givens rotation, etc.), we can construct a large class of transformed Hamiltonians H' . As an important bonus, the accelerated convergence of perturbation theory provides an efficient method for performing the calculations. This allows for the exploration of a very large class of high-fidelity chiral interactions with a wide range of different short-distance regulators.”

While discussing the topic of conceptual advances contained in this work, we should also mention the novel strategy used to push beyond the current bottleneck that is thwarting progress in *ab initio* nuclear structure theory. While the *ab initio* nuclear theory community is not very large in number, the question of whether *ab initio* nuclear theory will be reliable and accurate enough to address key questions of relevance to new experiments exploring new emergent phenomena and physics beyond the Standard Model is of extremely broad scientific interest. In the revised main text, we now write:

“The results in Ref.^{28,29} showed that the range and locality of the nuclear interactions have a strong influence on nuclear binding. The ${}^4\text{He}$ nucleus, also called an α particle, is a spatially compact object whose radius is comparable to the range of the interactions between nucleons. The central density of the α particle is about 50% greater than the saturation density of nuclear matter. As a result, the $\alpha\alpha$ interaction is highly sensitive to the range and locality of the nucleonic interactions. These same arguments apply to other interactions involving α particles and nucleons, which we denote as N . Using the formalism of cluster effective field theory^{30–33} for α particles and nucleons, the two-cluster interactions are $\alpha\alpha$ and αN , and the three-cluster interactions are $\alpha\alpha\alpha$, $\alpha\alpha N$, and αNN , with αNN having two possible isospin channels. In addition to the $\alpha\alpha$ interaction, there may be some dependence on short-distance physics arising from these other cluster interactions. This gives us a simple counting argument that there is one important parameter associated with the $\alpha\alpha$ interaction and at most five other parameters that also require

tuning to reduce unwanted errors. Our strategy is to tune the short-distance features of the three-nucleon interactions to achieve this error cancelation.”

Below, is a list of comments (not in order of relevance) that I believe should be addressed by the Authors.

1.) As the Authors point out, the long-standing challenge of accurately reproducing energies, radii, and saturation has received a lot of attention. It is still not clear what makes some interactions give reliable results for heavier systems and others not. This is an important question and we certainly need a much better understanding of the nuclear interaction, and new takes on the problem are important. However, there are results (e.g. Refs. 19,56) based on NN+3NF interactions in chiral EFT up to third order (NNLO) that describe energies and radii of selected nuclei up to 208Pb and nuclear matter saturation consistent with experimental data, without the need for additional short-range terms. Those calculations also employ a regulator cutoff similar to the one imposed by the lattice EFT spacing in the present work. The fourth order (N3LO) corrections, and somehow beyond, as is done in the manuscript, should slightly improve predictions. Judging from Fig. S4 there appears to be a nearly 50% correction in the 40Ca energy coming from the higher order short-range terms. Why not re-fit the standard 3N terms c_D and c_E to some heavy data in wave 1? What are the new insights and what is gained by adding more 3N contacts? Does the approach qualify as an EFT and how to systematically improve?

We thank the referee for this excellent question and discussion. As mentioned in our reply to Referee 1, Jiang et al., Phys. Rev. C 102, 054301 (2020) and Hu et al., Nature Phys. 18, 1196 (2022), are both excellent papers. Hu et al. is groundbreaking in extending the reach of *ab initio* methods to ^{208}Pb . The area that needs further improvement is mentioned in our reply to Referee 1. As discussed in Nosyk et al., Phys. Rev. C 104, 054001 (2021), the $^3\text{S}_1$ and $^1\text{P}_1$ phase shifts in Jiang et al., are not accurate at relative momenta at 150 MeV and above. The typical momentum scale relevant for nuclear binding is about 150 MeV, and so this seems to be less than optimal. Nosyk et al. found that putting the phase shifts at their empirical values creates tension with other observables such as nuclear saturation. We believe that this tension can be resolved by allowing the regulator structure of the three-nucleon interactions to be used as additional fit parameters rather than sacrificing the quality of the two-nucleon phase shifts. This suggestion has been privately conveyed to some of the authors of Jiang et al., and we hope that it might be helpful for future work.

If we perform WAVE 1 using $V_{c_D}^{(0)}$ and $V_{c_E}^{(0)}$ fitted to heavier nuclei, then the triton and alpha particles would be significantly overbound. This is a poor starting point, and we lose any hope of getting all of the low-energy physics correct by WAVE 5. Instead, we first need to get the properties of the lightest nuclei, ^3H and ^4He , correct in WAVE 1 and then maintain the accuracy for all successive waves. Our guiding principle is cluster effective field theory. We first need the individual clusters to have the correct physical properties. The underbinding for the heavier nuclei such as ^{40}Ca is primarily due to the $\alpha\alpha$ interaction being too weak. This comes in the next step. In the revised main text we have written:

“The results in Ref.^{28,29} showed that the range and locality of the nuclear interactions have a strong influence on nuclear binding. The ^4He nucleus, also called an α particle, is a spatially compact object whose radius is comparable to the range of the interactions between nucleons. The central density of the α particle is about 50% greater than the saturation density of nuclear matter. As a result, the $\alpha\alpha$ interaction

is highly sensitive to the range and locality of the nucleonic interactions. These same arguments apply to other interactions involving α particles and nucleons, which we denote as N . Using the formalism of cluster effective field theory^{30–33} for α particles and nucleons, the two-cluster interactions are $\alpha\alpha$ and αN , and the three-cluster interactions are $\alpha\alpha\alpha$, $\alpha\alpha N$, and αNN , with αNN having two possible isospin channels. In addition to the $\alpha\alpha$ interaction, there may be some dependence on short-distance physics arising from these other cluster interactions. This gives us a simple counting argument that there is one important parameter associated with the $\alpha\alpha$ interaction and at most five other parameters that also require tuning to reduce unwanted errors. Our strategy is to tune the short-distance features of the three-nucleon interactions to achieve this error cancelation.”

The $\alpha\alpha$ interaction produced by the two-nucleon interaction in H' is too weakly attractive. As explained in the new section “New Chiral Effective Field Theory Interactions with Wave Function Matching”, reducing the value of R slightly will increase the $\alpha\alpha$ attraction at the two-nucleon level without changing the phase shifts or the binding energies of ${}^3\text{H}$ and ${}^4\text{He}$. This alone would significantly improve the binding of ${}^{40}\text{Ca}$ at leading order. This also showcases the new capabilities of wave function matching to engineer new chiral effective field theory interactions. We write:

“As R is varied from 3.22 fm to 3.97 fm, the binding energy of ${}^{12}\text{C}$ decreases by 6 MeV while the binding energy ${}^{16}\text{O}$ decreases by 12 MeV. This is the physics of the quantum phase transition studied in Ref.⁷⁰”

The tuning of R will be explored in detail in future work. In this article, however, we focus on showing the general procedure for removing short-distance regulator errors in *ab initio* nuclear structure calculations. We want to present the general solution and how it can be implemented by other research groups using other *ab initio* methods.

In this work, we are able to reproduce accurate charge radii and nuclear saturation without fitting any parameters to charge radii and nuclear saturation. The NLEFT collaboration was previously able to get decent charge radii and nuclear saturation with a simple pionless effective field theory interaction with only four parameters in Ref.⁵⁰ Three parameters were used to tune the S-wave two-nucleon interaction and binding energy of ${}^3\text{H}$. Only one parameter, the strength of the local smearing, was used to reproduce the binding energies, charge radii, and all other many-body observables. This was possible due to the importance of the $\alpha\alpha$ interaction in nuclear binding.

In the new section “Properties of the Three-Nucleon Interactions” in Methods, we explain that the new three-nucleon interactions are simply changing the local regulator structure of the c_D and c_E three-nucleon terms:

“The three-nucleon interactions with regulator-dependent coefficients are illustrated in Fig. S14. Each of these interactions corresponds to the usual c_D and c_E three-nucleon terms in chiral effective field theory, but with different choices for the local smearing regulator. In essence, we are engineering the regulator structure of the c_D and c_E terms to cancel the dominant regulator-dependent errors that arise in *ab initio* nuclear structure calculations.”

In summary, we are using cluster effective field theory as a guiding principle to diagnose and fix systematic errors in nuclear many-body calculations using chiral effective field theory. As we can see from the 50

new predictions presented in the General Comments, the predictive power and accuracy appear to be very good. This is certainly an approach based on the principles of effective field theory. We are using one effective field theory to improve the accuracy and predictive power of another effective field theory. The procedure for systematic improvement at higher orders is clear. In order to increase the accuracy of the *ab initio* predictions, we increase the order for chiral effective field theory as well as the order for cluster effective field theory. Instead of just adding the binding energies of more and more nuclei to fit more parameters, cluster scattering observables should also be selected to target the properties of cluster effective field theory beyond leading order.

2.) *The Authors utilize the 3N short-range terms to remove unwanted dependence on short-range physics. However, the matching radius R is set to 3.7 fm, which is nearly three times the lattice spacing and on good grounds cannot be considered short range. Even 1.9 fm, which is also quoted, is of rather long range. How can this value of R be understood as short range?*

We thank the referee for this excellent question. As Referee 1 has asked a very similar question, we refer to the reply to Referee 1. A brief summary is that low-energy physics is independent of R . We give the theoretical argument why this happens, and then we present several benchmark examples confirming this fact numerically. The analysis is theoretically interesting and shows that wave function matching is completely different from renormalization group evolution.

3.) *For relative distances $r > 3.7$ fm, the unitary transformation U is not active so the Hamiltonian and the 2N wave functions are given by their high-fidelity versions, which at this range should be dominated by the leading-order one-pion exchange. For $r < 3.7$ fm, U is active and after matching the 2N wave functions are proportional, with a constant 1, to the leading-order and simple wave functions. A set of short-range 3N terms are then added and re-fitted to energies in heavy nuclei. This makes me wonder how different the results would be if one used only the leading-order 2N interaction plus a few (up to six or eight?) short-range 3N terms fitted to heavier systems instead of wave function matching? Related to this, there are also Lattice EFT calculations by Lu et al. (Physics Letters B 797 (2019) 134863) based pionless EFT NN+3N potentials with only 4 parameters, of which 2 are contacts, that yield results very similar to the ones in the present manuscript.*

This is a very interesting proposal. While it would not belong in this manuscript, we will consider investigating the topic and related questions in a future project. Let us try to speculate on the likely outcome.

As the referee mentions regarding Ref.,⁵⁰ the NLEFT collaboration was able to reproduce the bulk properties of light and medium-mass nuclei and neutron matter with decent accuracy. The success of this work can be understood as the four-parameter pionless effective field theory reproducing the low-energy properties of cluster effective field theory rather well. However, it does not accurately reproduce the low-energy phase shifts of the two-nucleon interaction. In particular, all of the spin-dependent interactions are completely omitted. This deficiency is the likely reason that some of the nuclear binding energies in Ref.⁵⁰ are not so accurate, as compared with the results presented in this work and the new results shown above in the General Comments.

By including some spin-dependent three-nucleon interactions, it may be possible to fix some of these problems. However, it seems unlikely that all significant deficiencies in the nuclear structure can be solved this way. We expect that the deficiencies to be especially difficult to repair for odd-odd nuclei with $N = Z$, where the two unpaired nucleons will have strong spin-dependent two-nucleon correlations such as tensor correlations.

4.) *As described in the Methods section, the Authors have carried out a careful history matching procedure complemented with a least-squares minimization to identify non-implausible and optimal LEC values. However, the number of unknown $3N$ short-range terms are comparable or even equal to the number of nuclear energies used for calibration in some waves of the history matching. Accounting for the fact that the leading-order couplings $c_{SU(4)}$ and c_I are allowed to vary appears to yield more parameters than data in waves no. 1/2. The situation is better in waves no. 4/5. Is this a problem? Are the Authors concerned about over-fitting in the least squares?*

We thank the referee for this question. The description in the original manuscript was causing this confusion. In the revised manuscript we now write:

“For the many-body calculations, we perform the following steps. We first write the transformed two-nucleon Hamiltonian H' as $H' = H^S + (H' - H^S)$. We then prepare eigenstates of H^S and apply corrections up to first order in perturbation theory to get the properties of the eigenstates of H' . The three-nucleon interactions are added to H' at this stage. In order to accelerate the convergence of perturbation theory further, we consider the more general partition $H' = H'^S + (H' - H'^S)$. The modified simple Hamiltonian H'^S has the same form as H^S in Eq. (S1), but we allow for different coupling strengths $c_{SU(4)}$ and c_I . We then minimize the energy to optimize the parameters. We should clarify that the parameters $c_{SU(4)}$ and c_I in H'^S are only used to improve the quality of the variational trial states and have nothing to do with the actual interaction parameters of the Hamiltonian.”

5.) *Figs. 2 and S4 nicely demonstrates the well-known (Phys. Rev. C 64, 014001, and Ref. 81) improvement of energy predictions for $A > 4$ nuclei when including some of them in the fit. In Fig. 2, the predicted nuclei with masses in $A=3-40$ are in essence interpolated since the interaction is anchored at $A=3$ and $A=40$ and in calibrated in between. It would add some credibility to the results if the Authors included energy (and radii) predictions beyond $40Ca$ using the same interaction.*

This is an excellent suggestion. Following the referee’s suggestion, we have performed new calculations for ^{50}Cr and ^{58}Ni using the same interactions. The results are presented in the revised main text as follows:

“We can also make predictions for heavier nuclei. For ^{50}Cr , we obtain 425(9) MeV for the binding energy and 3.64(7) fm for the charge radius. These compare well with the experimentally observed values^{37,38} of 435.05 MeV and 3.6588(65) fm. For ^{58}Ni , we obtain 493(6) MeV for the binding energy and 3.77(7) fm for the charge radius. These also compare well with the observed values^{37,38} of 506.46 MeV and 3.7757(20) fm.”

We note also the additional 50 new predictions presented in the General Comments above.

6.) *Fig. 3 shows predicted radii for $A=2-40$ and this is a very nice result. How well are the radii predicted for the other combinations of 3N contacts and interactions emerging from waves 1-5?*

We thank the referee for this excellent suggestion. We have performed these new calculations. In the new section in Methods called “Charge Radii”, we write:

“In Table S19 we show the computed charge radii of selected nuclei with fitted parameters and uncertainties determined using history matching for WAVE 1 through WAVE 5. For comparison, we also show the empirical values.”

- Besides the comments above, the abstract plus main text is well written and so are many parts of the Methods section. In some places clarification is needed and below is a list of minor comments to hopefully improve the manuscript.

A.) The value of the matching radius $R=3.7$ fm should be mentioned in the main text.

This is a very good point. In the revised main text we now write:

“For the calculations presented here, the value $R = 3.72$ fm is used. The dependence on R is discussed extensively in Methods.”

B.) Regarding the Nature Physics paper in Ref. 56; I consider this relevant also to the underlying physics discussion of the paper and not only the history matching part.

We thank the referee for the suggestion. Ref.²² is now cited along with the other *ab initio* calculations in the main text when discussing the underlying physics.

C.) It is not clear in the main text, or captions to Figs. 2 and 3, exactly which set of additional 3NF contacts are included and which of the many interactions listed in the Methods section that is used in the end. It can probably be deciphered from the Methods section, but it would help if this was pointed out in the main text.

This is another good suggestion. In the revised main text, we now write:

“We tune the short-distance features of the c_D and c_E three-nucleon interactions to minimize errors in the binding energies of selected light and medium-mass nuclei. A total of six additional three-nucleon parameters are adjusted, and in Methods we present the details of these parameters along with a detailed description of the fitting procedure and the resulting uncertainty.”

D.) Are the results in Figs. 2 and 3. point-estimates or posterior predictive distributions? Also, it is not easy to find the lattice predictions for the pure neutron matter energies in Fig. 3.

We thank the referee for the question and suggestion. The binding energy results shown in the right panel of Fig. 2 are evaluated from the posterior predictive distributions computed using history matching. The charge radius results in the left panel of Fig. 3 as well as the pure neutron matter and symmetric nuclear matter results in the right panel of Fig. 3 are all point estimates. In the new section “Charge Radii” of Methods, we show the calculated charge radii of selected nuclei at with fitted parameters and uncertainties determined using history matching for WAVE 1 through WAVE 5. These are posterior predictive distributions computed using history matching.

In the revised main text, we write for the binding energy results:

“The one-standard-deviation posterior distribution error bars shown in Fig. 2 represent uncertainties due to Monte Carlo errors, infinite-volume extrapolations, infinite projection time extrapolations, and uncertainties associated with the interactions.”

For the charge radii results, we write:

“The one-standard-deviation point estimate error bars shown in Fig. 3 represent computational uncertainties due to Monte Carlo errors, infinite volume extrapolation, and infinite time extrapolation.”

For the neutron matter and nuclear matter results, we write:

“The one-standard-deviation point estimate error bars shown represent computational uncertainties due to Monte Carlo errors and infinite projection time extrapolation.”

For the new table in “Charge Radii” in Methods, we write:

“In Table S19 we show the computed charge radii of selected nuclei with fitted parameters and uncertainties determined using history matching for WAVE 1 through WAVE 5. For comparison, we also show the empirical values. The one-standard-deviation posterior distribution error bars include Monte Carlo errors, infinite-volume extrapolations, infinite projection time extrapolations, and uncertainties associated with the interactions.”

The right panel of Fig. 3 has been updated, and the results for pure neutron matter energies are plotted using the same plot symbols as for symmetric nuclear matter. The details regarding box sizes and particle numbers are given in the text.

E.) Systematic theoretical errors are estimated to be 0.1 MeV per nucleon. I could not find a clear explanation of this number in the Methods section.

We thank the referee for suggesting this clarification. In the main text we now write:

“As a first test, we consider the energy of the deuteron, ${}^2\text{H}$. The wave function matching calculation gives a binding energy of 2.02 MeV, in comparison with 2.21 MeV for the true binding energy of H and 2.22 MeV for the experimentally observed value. The residual error of 0.1 MeV per nucleon is due to corrections beyond first order in powers of $H' - H^S$.”

Later we write:

“The good agreement with the Tjon band suggests a residual error of 0.1 MeV per nucleon or less for ^3H and ^4He . In Methods we present numerical evidence that the estimate of 0.1 MeV error per nucleon is also valid for light and medium-mass nuclei.

And then we say:

“As described in Methods, we estimate the additional systematic errors due to truncation of the expansion in powers of $H' - H^S$ to be approximately 0.1 MeV per nucleon.”

In Methods we explain the evidence and reasoning:

“As we include more three-nucleon interactions in Table S6, Table S7, Table S8, Table S9, and Table S10, the descriptions of the energies differences get better. The best root-mean-square error for the binding energy per nucleon drops to 0.29 MeV, 0.13 MeV, 0.11 MeV, 0.10 MeV, and 0.08 MeV respectively. We see that, after including three additional three-nucleon interactions, we reach a plateau of about 0.1 MeV error in the binding energy per nucleon. From this analysis, we conclude there are three or four independent parameters that must be tuned in order to remove significant regulator-dependent artifacts from the *ab initio* calculations. This conclusion can also be drawn from the history matching analysis. In Fig. S4, we see that there is only minor improvement in the size of the estimated errors and the accuracy of the predictions when going beyond WAVE 3.

The error plateau of 0.1 MeV energy per nucleon indicates that there are some additional sources of error at this level. For most calculations, the Monte Carlo error and Euclidean time extrapolation is about a factor of two smaller than this amount. The 0.1 MeV energy per nucleon is more consistent with the size of the errors from higher corrections in perturbation theory. We can estimate this error by varying the interaction coupling for the Hamiltonian used to prepare the nuclear many-body wave function and noting the variation in the expectation value of the N3LO Hamiltonian H' with three-nucleon interactions included. We note that first-order perturbation theory for the energy is equivalent to a variational calculation for the energy. In Table S18, we present results for several example nuclei at $L_t = 741$ lattice time steps, or Euclidean time duration 0.741 MeV^{-1} . From the variations in the expectation value of H' , we see that an error estimate of about 0.1 MeV per nucleon is a reasonable estimate of the size of the higher-order perturbation theory corrections to the energy.”

F.) It would be useful to know how the numerical values of the smearing parameters below Eq. S7 are determined and whether this would impact the results?

This is an excellent question. These parameters were intentionally chosen to be similar to the parameters used in Lu et al., Phys. Lett. B 797, 134863 (2019) and Lu et al., Phys. Rev. Lett. 125 192502 (2020). There is a discussion of the physics of the smearing parameters in the Supplemental Material of Lu et al., Phys. Lett. B 797, 134863 (2019). In the revised text, we now write:

“Throughout our calculations we use local smearing parameter $s_L = 0.07$ and nonlocal smearing parameter $s_{NL} = 0.5$. These parameters are similar to the values used in Ref. 47 and Ref. 4 for the same

lattice spacing. Both s_L and s_{NL} contribute to the range of the two-nucleon interaction. However, s_L has a special role because the local smearing has a large impact on the $\alpha\alpha$ interaction, which in turn is important for nuclear binding. The values of these parameters are essential for dictating the many-body properties of the simple Hamiltonian H^S . The success of wave function matching for the many-body system relies on H^S having the correct basic features of the many-body system of interest, albeit with lower fidelity.”

G.) Maybe it is implicitly defined somewhere in the Methods section, but I could not see a clear definition of, or reference to, the modified SU(4) short-range 3N terms. For instance, where is $V_{CD}^{(2)}$ defined.

These 3N forces are defined in Eq. (S16), (S18), (S17), and (S19), and the short-hand notations for the density operators used in the definition of these forces are given in Eq. (S10) and (S11). In the manuscript we write:

“For this term we consider the SU(4) symmetric three-nucleon forces $V_{CE}^{(0)}$, $V_{CE}^{(1)}$, $V_{CE}^{(2)}$, $V_{CE}^{(l)}$, $V_{CE}^{(t)}$ given in Eqs. (S16), (S18), (S19), and the one-pion-exchange three-nucleon forces $V_{CD}^{(0)}$, $V_{CD}^{(1)}$ and $V_{CD}^{(2)}$ defined in Eq. (S17).”

H.) The notation \mathbf{x} in connection with the emulator in Eq. S26 and minimization in Eq. S27 is confusing. Immediately below Eq. S26, the elements of this vector are declared to denote only(?) the NN LECs, i.e., the x_i 's in the first term of Eq. S26. After this, it becomes a bit muddy how the β_k 's enter the history matching analysis. For instance, it looks like only the NN LECs are fitted using least squares. Which I presume is not the case.

We thank the referee for pointing about the lack of clarity in the original manuscript. In Methods we now clarify the process as follows:

“The first step in our analysis is to use the theoretical errors shown in Fig. S2 and get posterior distributions of the 2N LECs sampling with Markov Chain Monte Carlo (MCMC), and the results are shown in Fig. S3. For each interaction, we approximately collect 10^3 samples. Then we can use 2N LECs from these distributions as inputs, which form a list represented by a vector \mathbf{x} where the individual inputs are x_i with $i = 1, \dots, d$. We define our *ab initio* model for any number of outputs and write $f_o(\mathbf{x}, \beta)$ for individual outputs $o = 1, 2, \dots, q$ and adjustable parameters labeled by the vector β . We define the following emulator equation for our *ab initio* model,

$$f_o(\mathbf{x}, \beta) = \sum_j \mathbf{x}_j \left(\frac{\partial E}{\partial \mathbf{x}} \right)_{jo} + \sum_k \beta_k \left(\frac{\partial E}{\partial \beta} \right)_{ko} . \quad (3)$$

The first term on the right hand side is the model output of only 2N interactions. This term uses 2N inputs from the distributions shown in Fig. S3, which are denoted by the vector \mathbf{x}_j , where the subscript j denotes different 2N interactions. The second term on the right hand side is the output of only 3N interactions, and it is a calibration term where β_k are unknown regression coefficients which will correspond to the 3N LECs at the end of the analysis. For this term we consider the SU(4) symmetric three-nucleon forces $V_{CE}^{(0)}$, $V_{CE}^{(1)}$, $V_{CE}^{(2)}$, $V_{CE}^{(l)}$, $V_{CE}^{(t)}$ given in Eqs. (S16), (S18), (S19), and the one-pion-exchange three-nucleon forces $V_{CD}^{(0)}$, $V_{CD}^{(1)}$ and $V_{CD}^{(2)}$ defined in Eq. (S17). In the emulator Eq. (S23), the derivatives of the energies with

respect to x and β stand for the operators calculated on the lattice using first-order perturbation theory. In the emulator equation all inputs are active variables, as so we do not make any such distinction. The emulator works for any number of outputs, $f_o(\mathbf{x}, \beta)$ with $o = 1, 2, \dots, q$. When we emulate our model's behavior for x_i inputs, with $i = 1, \dots, d$, we use the experimental data corresponding to the outputs and the calibration coefficients β_k so that there is acceptable agreement between our model and the experimental data. Therefore, the calibration coefficients β_k are obtained from a nonlinear least-squares fitting by minimizing the expression for each input set in the vector \mathbf{x} ,

$$r_n(\mathbf{x}) = \left[\min (f_n(\mathbf{x}, \beta) - z_n)^2 \right]^{1/2}, \quad (4)$$

where z_n is the experimental data.”

I.) How is the natural size for the 3N LECs β_k estimated? The inference is conditional on this information and it should be quantified.

We thank the referee for this question. In the revised “Uncertainty Analysis” in Methods, we write:

“When we evaluate Eq. (S24), we impose a constraint on the least squares problem so that only natural sized parameters are allowed for the coefficient β_k . Due to the correspondence of these coefficients β_k to the 3N LECs, their values are restricted to ensure that the expectation values of individual 3N interactions do not exceed 30% of the total contribution from the expectation value of the 2N interactions. In our numerical experiments, we observed a significant adverse impact on the results for the excited states in the absence of such a constraint.”

J.) Figure S3 should be equipped with numerical values on the x-axis to provide a scale for the posterior distribution of the NN LECs. The prior for the NN LECs should also be specified.

We thank the referee for this suggestion and the many other helpful comments, questions, and suggestions. Figure S3 is now updated. The new version shows the horizontal scale markings for the posterior distribution of the NN LECs in lattice units. In this updated figure, the solid-red dashed lines show the prior distribution of the LECs which give the N3LO phase shifts with the theoretical error bands in Figure S2.

Reviewer Reports on the First Revision:

Referees' comments:

Referee #1 (Remarks to the Author):

We appreciate the authors' response and considerable additional material. A main point remains that the radius of the wave function matching certainly is not a short distance. The corresponding momentum is about a factor three lower than the lattice momentum cutoff and is close to the Fermi momentum in nuclear matter. At nuclear saturation density the distance between alpha particles is about 2.5fm, and the average distance between nucleons is about 1.8fm. From this perspective the matching radius of 3.72fm is not a short distance, and this is wrongly stated in the abstract.

The new T_{jon} -line results in figure S5 are inconclusive. Not enough points between $R=3.22\text{fm}$ and 2.63fm are shown, and ideally one would like to go closer to the minimum distance of 1.86fm . While the authors seem to attribute deviations from the T_{jon} line to the failure of perturbation theory, it could as well be due to R not being a short distance.

It seems that the wave function matching and the large number of three-body contacts serve to introduce a medium-distance scale such that a sign problem is avoided, and a good amount of nuclear data can be fit. The resulting Hamiltonian is tailored to the distance R . As the authors write, changing R from 3.22 to 3.97 yields changes in ^{12}C and ^{16}O of about 0.5 and 0.75 MeV per nucleon, respectively.

Referee #2 (Remarks to the Author):

Dear Editor,

I wish to express my sincere appreciation to the authors of the manuscript for their diligence in addressing my comments and those of the other referees. Their dedication to improving their work is commendable. However, I must share my disappointment with the revisions made.

I find it challenging to fully grasp the rationale behind the absence of a benchmark comparison of this new method with other ab initio calculations. In their response, the authors assert that other methods "do not use the same starting Hamiltonian." Are not all these methods (No-Core Shell Model, Green's function Monte Carlo etc.) supposed to be ab initio? It therefore seems reasonable to expect convergence towards similar solutions when rooted in the same realistic nuclear interactions. I believe that introducing a new, presumably more efficient, approximate method should ideally entail a comprehensive comparison with established, non-approximate methods. The tests presented in the Methods section appear overly simplified. Would it not be beneficial to compare the wave-function

matching approach to calculations using realistic interactions among nucleons with finite masses? Some of the co-authors of this paper have previously conducted ab initio calculations on nuclei like ^{12}C or ^{16}O (as referenced in [71-73]). Could the new method not be compared, at the very least, to these earlier lattice calculations employing the exact same nuclear interaction?

I remain puzzled by the authors' response to my query regarding the use of Cluster-EFT within their calculation framework in the third paragraph of page 3 of the manuscript. The explanation provided still appears somewhat cryptic. In their response, they mention that this is an A-body calculation. However, the inclusion of α - α , α -N, α - α - α ... interactions raises questions. If this is indeed an ab initio calculation, why is there a need to account for cluster formation within the nucleus? Additionally, I note that the manuscript does not explicitly specify that the work corresponds to an A-body calculation. While it may be implicit when describing it as ab initio, the term "ab initio" can carry different connotations across fields, such as in lattice QCD, nuclear physics, or atomic physics.

Actually, most of these issues point towards the major flaw of this manuscript: its lack of clarity. The five-page text lacks a clear structure, making it challenging to follow. For instance, the paragraph on Cluster EFT is sandwiched between discussions on regulating the short-range features of the nuclear interaction and the necessity for three-nucleon forces. Is this the most logical sequence, and is this piece of information indispensable for understanding wave function matching? Perhaps it would be more appropriate to place it towards the end of the Methods section.

Regrettably, the Methods section also lacks clarity. The various sections do not form a cohesive narrative, and their order appears arbitrary. While the manuscript centers on the novel "wave function matching" technique, the Methods section commences with an extensive Sec. S2, spanning nearly eight pages, detailing the fitting of the nuclear interaction. While I acknowledge the importance of precise nuclear interactions for such calculations, it raises the question of whether this constitutes the core of the authors' work. The scant details about the method itself are relegated to Secs. S5 and S6, appearing on pages 18 and 19 of the manuscript.

It remains unclear whether the fitting described in Sec. 2 is conducted using the wave-function matching approach or within a more conventional lattice method. If it is the former, there is potential concern that the new nuclear interactions may be influenced by the method itself. Given the approximations inherent in the wave-function matching, there may be doubts about the integrity of parameters, particularly those related to the three-body force. How can we be sure that these parameters are not affected by the three-body forces induced by the method? This underscores the necessity for a clear benchmark.

Scientific publishing serves the purpose of disseminating novel findings to the community. Due to the wide readership of Nature, clarity and precision in communication are paramount. Regrettably, the present manuscript lacks both of these qualities. While I acknowledge the potential significance of the results, the lack of a coherent structure in the authors' presentation and their reluctance to conduct suggested tests lead me to recommend, with regret, against accepting this manuscript for publication.

Referee #3 (Remarks to the Author):

I sincerely thank the Authors for their great efforts to improve the manuscript and responding to most of my comments. The new detailed discussion and analysis about the distance "R" testifies to the usefulness and validity of wave function matching. The Authors also make a compelling case for adjusting short-range three-nucleon interactions based on cluster EFT principles to approximate crucial cluster physics. I believe this research is top-tier and innovative in the field of physics. However, I'm not convinced that wave function matching and the current findings about EOS predictions and radii/energies for medium-mass nuclei hold exceptional scientific significance for a broad interdisciplinary audience.

I have concerns about the assertion of predictive power, the way uncertainties are presented, and the methods for enhancing the results/predictions systematically. This is tied to the opening comments and (primarily) questions 1), 6), H), and J) in my previous report.

-- about predictive power

I am trying to understand how the quality of the predictions for radii evolve wave-by-wave in the history matching analysis. The interaction is modified and refit in each wave, which is excellent use of history matching. As pointed out by the Authors, there is little improvement in energies after WAVE 3, and the corresponding interaction is more or less as accurate (for energies) as WAVE 5, and that is a very good result (for energies). However, the significance of the fits in WAVES 4-5 for radii remains unclear, which is crucial for comprehending the Authors' analysis of nuclear saturation.

The Authors now include wave-by-wave data for radii in table S19 in the updated methods section. This is a very valuable addition. It would be even more valuable to see those results like the energies in Fig. S4. The table suggests that the radii of heavier nuclei, like ^{28}Si and ^{40}Ca , improve when fitting energies for heavier nuclei. Does this imply that energy fitting in heavy-mass nuclei is necessary for obtaining accurate radius predictions for such nuclei?

The new predictions for the energies and radii of ^{50}Cr and ^{58}Ni goes hand in hand with the EOS predictions and are certainly a valuable addition to the manuscript. Why not add these results to Figs 2-3 and S4?. It would also be valuable to add the energy and radius predictions for ^{50}Cr and ^{58}Ni wave-by-wave.

The text in the methods section states that table S19 consists of computed values based on interactions from history matching. It is not clear why the values in the columns labelled 'Rc' and 'WAVE 5' differ. Also, is the notation for the charge radius (Rc) consistent with the notation in Eq. S50?

Predictions for even heavier nuclei ($A > 100$ or beyond) would be greatly appreciated. I realize the (prohibitive?) computational expense of doing such calculations, and I am not asking the Authors to undertake such calculations now. But it would likely help the reader understand the context of the results if the Authors commented on the prospects of extending their predictions to heavier nuclei.

-- about systematic improvement and truncation errors

There is a discussion in the response (p.24) about how one effective field theory is used to improve the accuracy and predictive power of another effective field theory. This is very good use of EFTs! I agree that the procedure for systematic improvement is clear, in principle.

The main results for energies, radii, and EOS build on (chiral) EFT and wave function matching uses interactions truncated at LO and N3LO. At the moment, the Authors build their case of predictive power based solely on comparison with known data.

I'm somewhat surprised that the Authors haven't addressed the EFT truncation errors, which are likely present. My question is: what is the fidelity of the predictions based on the "high-fidelity" interactions discussed in the manuscript? This is important when making claims about predictive power in cases where no data exist. I think it would be valuable to add estimates of the truncation error to the other important sources of uncertainty in Figs. 2-3.

In section S7, the Authors state that wave function matching "defines a new low-energy effective field theory with the same breakdown scale as the original low-energy effective field theory." A similar

statement is made in the reply to ref 2: "This benchmark example provides numerical confirmation that, when applying wave function matching to chiral effective field theory, the transformed Hamiltonian H' can be viewed as a new chiral effective field Hamiltonian." (p. 13, reply to ref 2).

Wave functions change in "wave function matching", but in what way is the new chiral effective field theory interaction H' different from the starting LO and N3LO interactions? Is it correct to say that the LO NN interaction receives a short-range modification that facilitates perturbative N3LO calculations?

-- about the uncertainty analysis

The concept of an emulator is introduced in Eq. S23. A discussion about the emulator uncertainty would be good to include here, or at least referred to.

If (only) β is minimized in Eq. S24 I think it helps the reader to indicate this below the \min in Eq. S24.

It is unclear what is meant by "prediction error" below Eq. S25. Doesn't the variance of " r " account for part of a prediction error already? What is the relation between data, emulator, model, ... and their uncertainties?

Where can I find the estimate used for $\text{Var}(\epsilon)$?

How is the likelihood that connects the prior and posterior in Fig S3 defined? Is it a diagonal normal distribution or something else?

In the caption of Fig. S3, it says that the prior (solid red line) gives the theoretical error bands in Fig. S2. On p.9 it says that "[...] use the theoretical errors shown in Fig. S2 and get posterior distributions of [...] and the results are shown in Fig. S3.". I believe I understand how you obtain the posterior, but it seems like the posterior and prior are the same. Additionally, how was the prior determined?

Author Rebuttals to First Revision:

General Comments

We thank all of the referees for their help once again in improving this article. We have spent several months of work performing the suggested benchmark calculations, additional analyses, and improvements to the clarity of writing and structural ordering of the manuscript. Since the original submission of our manuscript in November 2022, we have done everything physically possible to make this article ready for publication in Nature. We are confident that this paper is now fully ready to be published in Nature.

Reply to Reviewer 1

We appreciate the authors' response and considerable additional material. A main point remains that the radius of the wave function matching certainly is not a short distance. The corresponding momentum is about a factor three lower than the lattice momentum cutoff and is close to the Fermi momentum in nuclear matter. At nuclear saturation density the distance between alpha particles is about 2.5fm, and the average distance between nucleons is about 1.8fm. From this perspective the matching radius of 3.72fm is not a short distance, and this is wrongly stated in the abstract.

We agree with the referee's suggestion. We have replaced the term "short distance" with "finite range" when referring to the wave function matching transformation. In the abstract, we now write:

"Wave function matching transforms the interaction between particles so that the wave functions up to some finite range match that of an easily computable interaction."

Later in the article, we write:

"While keeping the observable physics unchanged, wave function matching creates a new high-fidelity Hamiltonian H' such that wave functions up to some finite range match that of a simple Hamiltonian H^S which is easily computed."

And then again later:

"We define a finite-range projection operator P_R that projects out the portion of the two-nucleon state with separation distance less than or equal to R . We let $|m\rangle$ for $m = 1, \dots, m_R$ be an orthogonal basis spanning the set of two-nucleon channel states up to that finite range so that $P_R = \sum_{m=1}^{m_R} |m\rangle \langle m|$."

The new Tjon-line results in figure S5 are inconclusive. Not enough points between $R=3.22$ fm and 2.63 fm are shown, and ideally one would like to go closer to the minimum distance of 1.86 fm. While the authors seem to attribute deviations from the Tjon line to the failure of perturbation theory, it could as well be due to R not being a short distance.

We thank the referee for the comment and suggestion. As R specifies radial distance on the lattice, it takes discrete values $(n_x^2 + n_y^2 + n_z^2)^{1/2}a$, where n_x, n_y, n_z are integers and a is the lattice spacing. For lattice spacing $a = 1.32$ fm, the allowed radial distances for R are 0 fm, 1.32 fm, 1.86 fm, 2.28 fm,

2.63 fm, 2.94 fm, 3.22 fm, 3.72 fm, etc. Following the referee’s suggestion, in the updated version of the manuscript, we have performed new calculations for the Tjon line, considering all possible values of R within the range 0 fm to 3.72 fm.

The updated results, depicted in Fig. S2, now offer a more thorough exploration of the Tjon line across the entire radial distance range. By including additional data, we address the concern raised by the referee about the interpretation and conclusiveness of the Tjon line results. We have updated the caption of Fig. S2 as well as the text in the relevant part of the manuscript as follows:

“In Fig. S2 we plot the energies of ${}^3\text{H}$ and ${}^4\text{He}$ for different values of the wave function matching radius R . We show lattice results at LO, NLO and N3LO in chiral effective field theory for all possible values of R less than or equal to 3.72 fm using two-nucleon interactions only. The corrections are calculated using first-order perturbation theory. The gray band is the predicted result from Ref.,¹ and the black open box shows the empirical point. We see that the results for $R > 2.7$ fm are largely independent of R . The deviation from the Tjon band for $R < 2.7$ fm is likely due to perturbation theory corrections beyond first order. The additional deviation for the LO results is due to the fact that Coulomb is not included. The fast convergence of perturbation theory in wave function matching requires that the radius R is somewhat larger than r_Δ . This appears to be satisfied for $R = 2.94$ fm, 3.22 fm and 3.72 fm, and the results are nearly identical for the values for $R > 2.7$ fm. Also, the condition for the fast convergence of perturbation theory is consistently met at LO, NLO, and N3LO in chiral effective field theory as seen in Fig. S2.”

Regarding the deviation from the Tjon line, it is worth noting that the issue of convergence of perturbation theory for the Tjon line has been demonstrated for lattice calculations in Ref.⁵ As shown in Table I of Ref.⁵ the ratio B_4/B_3 evolves from 4.38 in first-order perturbative calculations to 3.49 in second-order perturbative calculations. This is in good agreement with the results presented here. By accelerating the convergence of perturbation theory, wave function matching enables us to solve the problem efficiently at first order in the perturbation theory. Therefore, in the revised version of the manuscript we present some new results to address the referee’s criticism regarding the connection between the deviation from the Tjon line and the failure of perturbation theory. The new results are shown in Table S1 and in the relevant section we added new text as follows:

“To further elucidate the above discussion, we assert that the slow convergence of perturbation theory not only causes deviations from the Tjon line but also disrupts the convergence of chiral effective field theory. This observation is exemplified by the binding energies of ${}^2\text{H}$, ${}^3\text{H}$, and ${}^4\text{He}$ presented in Table S1 for varying values of R in the range of 0 fm to 3.72 fm using two-nucleon interactions only. The outcomes clearly illustrate that when first-order perturbation theory inadequately approximates the solution, the convergence of chiral effective field theory becomes less controlled. Nevertheless, a remarkable resolution to this challenge emerges, as demonstrated by the results in Table S1. The rapid convergence of perturbation theory through the use of wave function matching elegantly rectifies the problem.”

The alternative explanation that the deviation from the Tjon line is caused by R not being a short distance requires that the deviation from the Tjon line diminishes for small R . This is the opposite of what is observed.

It seems that the wave function matching and the large number of three-body contacts serve to introduce a medium-distance scale such that a sign problem is avoided, and a good amount of nuclear data can be fit. The resulting Hamiltonian is tailored to the distance R . As the authors write, changing R from 3.22 to 3.97 yields changes in ^{12}C and ^{16}O of about 0.5 and 0.75 MeV per nucleon, respectively.

We thank the referee for the insightful comment. This is a fair assessment of our work.

Reply to Reviewer 2

I wish to express my sincere appreciation to the authors of the manuscript for their diligence in addressing my comments and those of the other referees. Their dedication to improving their work is commendable. However, I must share my disappointment with the revisions made.

*I find it challenging to fully grasp the rationale behind the absence of a benchmark comparison of this new method with other *ab initio* calculations. In their response, the authors assert that other methods "do not use the same starting Hamiltonian." Are not all these methods (No-Core Shell Model, Green's function Monte Carlo etc.) supposed to be *ab initio*? It therefore seems reasonable to expect convergence towards similar solutions when rooted in the same realistic nuclear interactions. I believe that introducing a new, presumably more efficient, approximate method should ideally entail a comprehensive comparison with established, non-approximate methods. The tests presented in the Methods section appear overly simplified. Would it not be beneficial to compare the wave-function matching approach to calculations using realistic interactions among nucleons with finite masses? Some of the co-authors of this paper have previously conducted *ab initio* calculations on nuclei like ^{12}C or ^{16}O (as referenced in [71-73]). Could the new method not be compared, at the very least, to these earlier lattice calculations employing the exact same nuclear interaction?*

We thank the referee for the comments and clarification. We now understand that the desired comparison is between different *ab initio* lattice calculations of nuclear structure with and without wave function matching. The references suggested by the referee are older calculations with a larger lattice spacing of 1.97 fm and systematic errors that have since been significantly improved. We have therefore chosen to perform a benchmark comparison based on the more recent N2LO lattice interactions described in Ref.⁶ We have added a new section called "Benchmark *ab initio* calculations with and without wave function matching" in Methods as follows:

"In the following, we perform *ab initio* calculations of ^3H , ^4He , ^8Be and ^{12}C with and without wave function matching. We use the N2LO lattice interactions described in Ref.,⁶ which was used to compute the energies and intrinsic structures of the low-lying states of ^{12}C . We use a spatial lattice spacing of $a = 1.64$ fm. The N2LO chiral interaction has two-nucleon interactions of the form ...

In Table S4, we show the N2LO results obtained with and without wave function matching. We use spatial periodic boxes of lengths $L = 19.7, 16.4, 16.4,$ and 13.1 fm for the calculations of ^3H , ^4He , ^8Be and ^{12}C , respectively, and we perform the Euclidean time extrapolation to infinity. We present results for $E_{2\text{NF}_s}$ (two-nucleon interactions only), $\Delta E_{3\text{NF}_s}$ (corrections from three-nucleon interactions) and $E_{2\text{NF}_s+3\text{NF}_s}$ (total). The excellent agreement for $E_{2\text{NF}_s+3\text{NF}_s}$ with and without wave function matching shows strong evidence that wave function matching is not changing the low-energy physics of these nuclei. This is a

non-trivial check since the results at $E_{2\text{NF}_s}$ are different. Wave function matching is producing an induced three-nucleon interaction, but this induced three-nucleon interaction is consistent with a renormalization of the coefficients of the N2LO chiral three-nucleon interactions. Due to the softness of these interactions, there appears to be little impact of the wave function matching transformation beyond these induced three-nucleon interactions. We note that the two sets of calculations are performed with the same auxiliary field configurations, and so the computational errors for the two calculations are correlated.”

I remain puzzled by the authors’ response to my query regarding the use of Cluster-EFT within their calculation framework in the third paragraph of page 3 of the manuscript. The explanation provided still appears somewhat cryptic. In their response, they mention that this is an A-body calculation. However, the inclusion of alpha-alpha, alpha-N, alpha-alpha-alpha... interactions raises questions. If this is indeed an ab initio calculation, why is there a need to account for cluster formation within the nucleus? Additionally, I note that the manuscript does not explicitly specify that the work corresponds to an A-body calculation. While it may be implicit when describing it as ab initio, the term ”ab initio” can carry different connotations across fields, such as in lattice QCD, nuclear physics, or atomic physics.

We thank the referee for requesting the clarification. We have now made the language as direct and explanatory as possible on page 3 of the main text:

“Our strategy is to tune the short-distance features of the three-nucleon interactions to achieve this error cancellation. The binding energies of various light nuclei with cluster substructures allow us probe the relationship between the three-nucleon interactions and the effective cluster interactions. We should emphasize that our calculations are full A-body calculations, and cluster effective field theory is only used to diagnose sensitivities to short-distance physics.”

Actually, most of these issues point towards the major flaw of this manuscript: its lack of clarity. The five-page text lacks a clear structure, making it challenging to follow. For instance, the paragraph on Cluster EFT is sandwiched between discussions on regulating the short-range features of the nuclear interaction and the necessity for three-nucleon forces. Is this the most logical sequence, and is this piece of information indispensable for understanding wave function matching? Perhaps it would be more appropriate to place it towards the end of the Methods section.

We thank the referee for this suggestion. The discussion of cluster effective field theory is now removed from the main text and is contained entirely in the Methods section. The main text now reads:

“In Methods, we use the formalism of cluster effective field theory^{30–33} for α particles and nucleons to provide a simple counting argument for the number of parameters that require tuning to reduce unwanted errors.”

Regrettably, the Methods section also lacks clarity. The various sections do not form a cohesive narrative, and their order appears arbitrary. While the manuscript centers on the novel "wave function matching" technique, the Methods section commences with an extensive Sec. S2, spanning nearly eight pages, detailing the fitting of the nuclear interaction. While I acknowledge the importance of precise nuclear interactions for such calculations, it raises the question of whether this constitutes the core of the authors' work. The scant details about the method itself are relegated to Secs. S5 and S6, appearing on pages 18 and 19 of the manuscript.

We thank the referee for this very helpful suggestion. The Methods section has now been reordered to form a coherent narrative, and we improved the clarity of each of the sections. The uncertainty analysis section has also been redone, fixing a deficiency in the Markov Chain Monte Carlo sampling of the prior distribution. The details of the changes are given in our response to Reviewer 3. We have also added preamble to Methods to help guide the reader. We write:

“The material in Methods is organized as follows. The first two sections are about nuclear lattice effective field theory and the lattice operators used. The next eight sections focus on wave function matching. We discuss theoretical concepts, benchmark tests, and the extension to continuous space. Then follows a section on uncertainties due to the chiral interactions. After this, we have four sections on aspects of chiral effective field theory, three-nucleon interactions, and new chiral interactions using wave function matching. We then conclude with a section on charge radii, a section presenting tables with details of the lattice results, and a section on three-nucleon interaction contributions to the binding energies.”

It remains unclear whether the fitting described in Sec. 2 is conducted using the wave-function matching approach or within a more conventional lattice method. If it is the former, there is potential concern that the new nuclear interactions may be influenced by the method itself. Given the approximations inherent in the wave-function matching, there may be doubts about the integrity of parameters, particularly those related to the three-body force. How can we be sure that these parameters are not affected by the three-body forces induced by the method? This underscores the necessity for a clear benchmark.

We appreciate the referee's thoughtful inquiry. Wave function matching does not change the nucleon-nucleon phase shifts, and so there is no difference whether the fitting of the phase shifts are done with the original high-fidelity Hamiltonian H or the transformed Hamiltonian H' . For the nuclear binding energies, however, we must use H' since wave function matching is essential for performing the required many-body calculations accurately. We agree that the new benchmark suggested by the referee and shown in Table S4 is helpful in establishing that the induced three-nucleon forces in wave function matching are indeed short-range interactions that can be absorbed into the fitted three-nucleon forces. These conclusions are also clearly established in the more pedagogical benchmark examples.

Scientific publishing serves the purpose of disseminating novel findings to the community. Due to the wide

readership of Nature, clarity and precision in communication are paramount. Regrettably, the present manuscript lacks both of these qualities. While I acknowledge the potential significance of the results, the lack of a coherent structure in the authors' presentation and their reluctance to conduct suggested tests lead me to recommend, with regret, against accepting this manuscript for publication.

We thank the referee for suggesting these important improvements to the manuscript. All of the suggested improvements have been implemented, and we believe that the new changes will help the broad audience of Nature readers to understand and appreciate the work.

Reply to Reviewer 3

I sincerely thank the Authors for their great efforts to improve the manuscript and responding to most of my comments. The new detailed discussion and analysis about the distance "R" testifies to the usefulness and validity of wave function matching. The Authors also make a compelling case for adjusting short-range three-nucleon interactions based on cluster EFT principles to approximate crucial cluster physics. I believe this research is top-tier and innovative in the field of physics. However, I'm not convinced that wave function matching and the current findings about EOS predictions and radii/energies for medium-mass nuclei hold exceptional scientific significance for a broad interdisciplinary audience.

I have concerns about the assertion of predictive power, the way uncertainties are presented, and the methods for enhancing the results/predictions systematically. This is tied to the opening comments and (primarily) questions I), 6), H), and J) in my previous report.

We thank the referee for the positive comments. In the following, we address point-by-point the remaining concerns of the referee.

– about predictive power

I am trying to understand how the quality of the predictions for radii evolve wave-by-wave in the history matching analysis. The interaction is modified and refit in each wave, which is excellent use of history matching. As pointed out by the Authors, there is little improvement in energies after WAVE 3, and the corresponding interaction is more or less as accurate (for energies) as WAVE 5, and that is a very good result (for energies). However, the significance of the fits in WAVES 4-5 for radii remains unclear, which is crucial for comprehending the Authors' analysis of nuclear saturation.

We thank the referee for the insightful question regarding the evolution of results wave-by-wave in our history matching analysis. We have completely redone the uncertainty analysis, fixing a deficiency in the Markov Chain Monte Carlo sampling of the prior distribution. The original history matching analysis

was sampling a larger distribution for the 2N LECs than intended. In the new history matching analysis, WAVES 4 and 5 have been combined into a single WAVE 4. The new results show that there is systematic improvement in the description of the charge radii with each successive wave, including at WAVE 4. We note that the charge radius calculations are not used at all in the history matching analysis and are therefore predictions. Nevertheless, we see systematic improvement in the charge radii as more data for the nuclear interactions are included. This provides evidence that the overall description of nuclear structure is improving and confirmation of the predictive power of our approach. In Methods, we now write:

“We note that the history matching analysis does not use any charge radii in determining the implausibility measures. Nevertheless, the overall description of the charge radii in Table S9 are improving with each successive wave as more features of the nuclear interaction are determined. When compared with the experimental values, the RMS deviations for the charge radii in Table S9 are 0.117(19) fm for WAVE 1, 0.097(10) fm for WAVE 2, 0.066(16) fm for WAVE 3, and 0.057(14) fm for WAVE 4. These estimates take into account the uncertainties of the history matching results in Table S9. If we use only the central values for the charge radii in Table S9, the corresponding RMS deviations are 0.113 fm for WAVE 1, 0.093 fm for WAVE 2, 0.053 fm for WAVE 3, and 0.033 fm for WAVE 4. We see that as more data for the nucleon-nucleon interaction and binding energies are included, the predictions of the charge radii systematically improve. This is exactly what we would like to see happen in *ab initio* calculations.”

The Authors now include wave-by-wave data for radii in table S19 in the updated methods section. This is a very valuable addition. It would be even more valuable to see those results like the energies in Fig. S4. The table suggests that the radii of heavier nuclei, like 28Si and 40Ca, improve when fitting energies for heavier nuclei. Does this imply that energy fitting in heavy-mass nuclei is necessary for obtaining accurate radius predictions for such nuclei?

We thank the referee for the useful suggestion. In the new version of the manuscript, Table S7 shows the nuclear binding energies for all the calculated nuclei with uncertainties from the history matching analysis for all four waves.

We agree that including some heavier nuclei is useful to detect any deficiencies in the nuclear interaction that gets systematically worse with increasing nucleon number. It is also useful to have heavier nuclei so that the corrections to the nuclear binding energies are bigger than the size of the computational errors. In the future, it will be useful to explore the many possible choices for the empirical data used for fitting parameters and the relative advantages and disadvantages.

The new predictions for the energies and radii of 50Cr and 58Ni goes hand in hand with the EOS predictions and are certainly a valuable addition to the manuscript. Why not add these results to Figs 2-3 and S4?. It would also be valuable to add the energy and radius predictions for 50Cr and 58Ni wave-by-wave.

We thank the referee for the useful suggestions. We are able to implement all of the suggestions that are physically possible given the computational resources available at present. We have added the binding energies of ^{50}Cr and ^{58}Ni to the right panel of Fig. 2 and the charge radii of ^{50}Cr and ^{58}Ni to the left panel of Fig. 3. The binding energies for ^{50}Cr and ^{58}Ni are now included in the history matching analysis, and the results are now shown in Fig. S11 and Table S7.

Extending the history matching analysis to the charge radii of ^{50}Cr and ^{58}Ni would require new and computationally expensive calculations. Unfortunately, the supercomputing resources available to us at this time are needed for other high-priority projects. We have spent one extra year performing new large-scale calculations associated with the revisions for this paper and reached the limit on the amount of supercomputing time that we can expend for this project. We hope that the referee will understand the real world constraints.

The text in the methods section states that table S19 consists of computed values based on interactions from history matching. It is not clear why the values in the columns labelled 'Rc' and 'WAVE 5' differ. Also, is the notation for the charge radius (Rc) consistent with the notation in Eq. S50?

We apologize for the confusing presentation of results appearing in the old Table S19 of the previous version of the manuscript. The original manuscript did not perform any history matching for the charge radii. Instead, the calculations were performed using the best fit values for the 2N LECs and 3N LECs. The history matching results for the charge radii of several nuclei were added later in the revised manuscript. These results were juxtaposed next to each other in the old Table S19, thereby causing some confusion.

In the new version of the manuscript, we are more clear about the results that are being presented. The results obtained using history matching are shown in Table S7 for the energies and in Table S9 for the charge radii. Meanwhile, the results obtained using the best fit parameters are presented in Table S10, along with an accounting of all the different errors. We write as follows:

“The central values presented are obtained using the best fit values for the 2N and 3N coefficients. The quoted errors include all the statistical and systematic uncertainties such as computational uncertainties from Monte Carlo errors, infinite volume extrapolation, and infinite projection time extrapolation. For the energies and several charge radii where the history matching analysis is performed, the uncertainties due to the interactions are also shown as a second set of quoted errors.”

We also thank the referee for pointing out the inconsistent notation between Eq. (S57) and Fig. 3. We have changed Eq. (S57) accordingly.

Predictions for even heavier nuclei ($A > 100$ or beyond) would be greatly appreciated. I realize the (prohibitive?) computational expense of doing such calculations, and I am not asking the Authors to undertake such calculations now. But it would likely help the reader understand the context of the results if the Authors commented on the prospects of extending their predictions to heavier nuclei.

We appreciate the referee's insightful comment and the recognition of the computational challenges associated with predictions for heavier nuclei ($A > 100$). We have added the following sentence into the manuscript:

“In this paper, we perform lattice Monte Carlo simulations of light nuclei, medium-mass nuclei up to $A = 58$, neutron matter up to $A = 80$, and nuclear matter up to $A = 160$. The method we employ can be used for calculations of heavier nuclei with $A > 58$ but may also benefit from increasing the efficiency of the computational algorithms and revisiting the parameterization of the three-body interactions.”

– about systematic improvement and truncation errors

There is a discussion in the response (p.24) about how one effective field theory is used to improve the accuracy and predictive power of another effective field theory. This is very good use of EFTs! I agree that the procedure for systematic improvement is clear, in principle.

We appreciate the referee's comment on our analysis and discussion.

The main results for energies, radii, and EOS build on (chiral) EFT and wave function matching uses interactions truncated at LO and N3LO. At the moment, the Authors build their case of predictive power based solely on comparison with known data.

I'm somewhat surprised that the Authors haven't addressed the EFT truncation errors, which are likely present. My question is: what is the fidelity of the predictions based on the "high-fidelity" interactions discussed in the manuscript? This is important when making claims about predictive power in cases where no data exist. I think it would be valuable to add estimates of the truncation error to the other important sources of uncertainty in Figs. S1-S10

We thank the referee for bringing attention to the issue of EFT truncation errors and their impact on our predictions. We note that Figs. S1 and S10 show the order-by-order convergence of chiral EFT at the two-nucleon level. We have now revised Fig. S10 to show the estimated chiral EFT truncation errors at NLO and N3LO. We are using a low-energy scheme where the two-nucleon interaction is the same at NLO and N2LO. We have also added the following material about EFT truncation errors:

“In our calculations we determine the LECs of the 2N chiral interactions at N3LO by fitting the calculated neutron-proton scattering phase shifts and mixing angles on the lattice to the Nijmegen PWA (NPWA).⁶⁷ The fundamental sources of uncertainties on these LECs are the systematic errors due to the truncated chiral EFT expansion and the statistical errors provided by the NPWA. In order to estimate the uncertainties, we follow the methodology introduced in Refs.^{84,85} The approach involves two components and in the first one, we define the systematic errors of an observable $X(p)$ due to the truncated chiral EFT expansion at order $N^m\text{LO}$ and momentum p using the following formula ...”

The chiral EFT truncation error at N3LO is used to define the prior distribution for the 2N LECs for the history matching analysis. When presenting the numerical results in Table S10, we now write:

“For the energies and several charge radii where the history matching analysis is performed, the uncertainties due to the interactions are also shown as a second set of quoted errors. These errors include propagated uncertainties due to truncation of the effective field theory expansion.”

In order to further address chiral EFT truncation errors, we have added a new table, Table S1. In Table S1, we show lattice results for the binding energies of ^2H , ^3H , and ^4He at LO, NLO and N3LO in chiral effective field theory for various values of the wave function matching range. The results clearly show that the convergence of chiral effective field theory is under control, and the order-by-order trend shows that the remaining EFT truncation errors at N3LO is smaller than the statistical errors for these light nuclei. For heavier nuclei, the EFT truncation errors contribute to the sensitivity to short-distance physics that we are explicitly controlling with additional three-nucleon forces. In the revised text, we recognize the role of omitted higher-order interactions by writing:

“As a result, the $\alpha\alpha$ interaction is highly sensitive to the range and locality of the nucleonic interactions as well as omitted higher-order interactions.”

In section S7, the Authors state that wave function matching “defines a new low-energy effective field theory with the same breakdown scale as the original low-energy effective field theory.” A similar statement is made in the reply to ref 2: “This benchmark example provides numerical confirmation that, when applying wave function matching to chiral effective field theory, the transformed Hamiltonian H' can be viewed as a new chiral effective field Hamiltonian.” (p. 13, reply to ref 2).

Wave functions change in “wave function matching”, but in what way is the new chiral effective field theory interaction H' different from the starting LO and N3LO interactions? Is it correct to say that the LO NN interaction receives a short-range modification that facilitates perturbative N3LO calculations?

We appreciate the referee’s interesting and probing questions. In wave function matching, the transformed chiral Hamiltonian H' has the same phase shifts as the original chiral Hamiltonian H_{N3LO} . However the finite-range part of H'_{N3LO} is different from the finite-range part of H_{N3LO} . H'_{N3LO} is specially engineered so that the low-energy eigenstates of H'_{N3LO} , up to some finite range, look like the low-energy eigenstates of H^S . This makes perturbation theory converge quickly when starting from the low-energy eigenstates of H^S .

The referee refers to an LO interaction and an N3LO interaction. If the referee’s real intent is to ask about the simple Hamiltonian H^S and H'_{N3LO} , then the referee’s comments are almost correct. The only amendment is that H'_{N3LO} is the Hamiltonian that is modified to facilitate the fast convergence of perturbation theory. H^S is not modified at all.

If the referee is instead asking about the leading-order Hamiltonian and not H^S , then the following comments are relevant. There is no guarantee about the convergence of perturbation theory connecting $H'_{\text{LO}} = U^\dagger H_{\text{LO}} U$ and H'_{N3LO} or the convergence of perturbation theory connecting H^S and H'_{LO} . The special relationship produced by the transformation U is guaranteed only between H^S and H'_{N3LO} . If we wish to produce fast convergence of perturbation theory between some other pair of Hamiltonians, then we must construct the wave function transformation for those two Hamiltonians.

– about the uncertainty analysis

The concept of an emulator is introduced in Eq. S23. A discussion about the emulator uncertainty would be good to include here, or at least referred to.

We thank the referee for the comment, which provides an opportunity to improve our description. The revised text now reads:

“In the emulator, Eq. (S52), the derivatives with respect to $x_j^{(i)}$ and β_k stand for the derivatives of observables on the lattice using first-order perturbation theory. Therefore, the emulator uncertainty is the error due to the higher-order perturbations. As noted in Section S13, this is an additional source of systematic error that we estimate to be about 0.1 MeV energy per nucleon for the binding energies. However, as discussed in Section S13 and shown in Table S8, we are able to reduce this source of systematic error by allowing for variational optimization of the Hamiltonian used to prepare the nuclear many-body wave function. We perform this variational optimization so that the remaining systematic error is smaller than the estimated computational error due stochastic noise, Euclidean time extrapolation, and infinite volume extrapolation.”

If (only) β is minimized in Eq. S24 I think it helps the reader to indicate this below the min in Eq. S24.

We thank the referee for this suggestion. We have updated the Eq. (S54) as suggested.

It is unclear what is meant by “prediction error” below Eq. S25. Doesn’t the variance of “r” account for part of a prediction error already? What is the relation between data, emulator, model, ... and their uncertainties?

Where can I find the estimate used for $\text{Var}(\varepsilon)$?

We thank the referee for these questions that provide an opportunity to enhance the clarity of our manuscript. As noted above, we have redone the history matching analysis and the section has been rewritten to improve clarity. The text that caused the confusion is no longer in the revised manuscript. We now write:

“For notational convenience, we represent the 2N LECs as a vector \vec{x} with components x_j , indexed by j for each 2N LEC. Similarly, we represent the 3N LECs as a vector $\vec{\beta}$ with components β_k , indexed by k for each 3N LEC. We will work with sets of 2N LECs, which we collectively write as $\mathbf{x} = \{\vec{x}^{(i)}\}$. We will consider output observables z_o with index o . We write z_o^{exp} for the experimentally observed values and z_o^{theory} for the theoretical values that we compute using lattice simulations. We define our emulator for observable z_o as

$$f_o(\vec{x}^{(i)}, \vec{\beta}) = z_{o,\text{NP}}^{\text{theory}} + \sum_j x_j^{(i)} \frac{\partial z_o^{\text{theory}}}{\partial x_j^{(i)}} + \sum_k \beta_k \frac{\partial z_o^{\text{theory}}}{\partial \beta_k}.$$

The first term on the right-hand side, $z_{o,\text{NP}}^{\text{theory}}$, corresponds to the non-perturbative contribution to the observable as well as perturbative contributions from operators whose coefficients are not being varied in our history matching analysis. The second term on the right-hand side is the perturbative contribution from the 2N interactions. This term uses 2N inputs from the prior distributions shown in Fig. S12. The third term on the right-hand side is the perturbative contribution from the 3N interactions. For this term we consider the SU(4) symmetric three-nucleon forces $V_{CE}^{(0)}, V_{CE}^{(1)}, V_{CE}^{(2)}, V_{CE}^{(l)}, V_{CE}^{(t)}$ given in Eqs. (S16), (S18), (S19), and the one-pion-exchange three-nucleon forces $V_{CD}^{(0)}, V_{CD}^{(1)}$ and $V_{CD}^{(2)}$ defined in Eq. (S17). In the emulator, Eq. (S52), the derivatives with respect to $x_j^{(i)}$ and β_k stand for the derivatives of observables on the lattice using first-order perturbation theory. Therefore, the emulator uncertainty is the error due to the higher-order perturbations. As noted in Section S13, this is an additional source of systematic error that we estimate to be about 0.1 MeV energy per nucleon for the binding energies. However, as discussed in Section S13 and shown in Table S8, we are able to reduce this source of systematic error by allowing for variational optimization of the Hamiltonian used to prepare the nuclear many-body wave function. We perform this variational optimization so that the remaining systematic error is smaller than the estimated computational error due stochastic noise, Euclidean time extrapolation, and infinite volume extrapolation.

Let Z_w be the set of outputs that we consider in the w^{th} iteration of history matching, or wave index w . When we emulate our model’s behavior for inputs $\mathbf{x} = \{\vec{x}^{(i)}\}$, we use the corresponding experimental observables z_o^{exp} and calibrate the 3N LECs $\vec{\beta}$ so that there is acceptable agreement between our model and the experimental data. For each $\vec{x}^{(i)}$, we define $\vec{\beta}_*(\vec{x}^{(i)})$ to be the optimized $\vec{\beta}$ that achieves the least-squares fit for the relative error,

$$\sum_{o \in Z_w} \left[\frac{f_o(\vec{x}^{(i)}, \vec{\beta}_*(\vec{x}^{(i)})) - z_o^{\text{exp}}}{z_o^{\text{exp}}} \right]^2 = \min_{\vec{\beta}} \sum_{o \in Z_w} \left[\frac{f_o(\vec{x}^{(i)}, \vec{\beta}) - z_o^{\text{exp}}}{z_o^{\text{exp}}} \right]^2.$$

It is convenient to rewrite this using vectorized notation,

$$\sum_{o \in Z_w} \left[\frac{f_o(\mathbf{x}, \vec{\beta}_*(\mathbf{x})) - z_o^{\text{exp}}}{z_o^{\text{exp}}} \right]^2 = \min_{\vec{\beta}} \sum_{o \in Z_w} \left[\frac{f_o(\mathbf{x}, \vec{\beta}) - z_o^{\text{exp}}}{z_o^{\text{exp}}} \right]^2.$$

When we evaluate Eq. (S54), we impose a constraint on the least squares problem so that only natural sized parameters are allowed for the 3N LECs $\vec{\beta}$. The values are restricted to ensure that the expectation values of individual 3N interactions do not exceed 30% of the total contribution from the expectation value of the 2N interactions. In our numerical experiments, we observed a significant adverse impact on the results for the excited states in the absence of such a constraint.

Let G_w denote the volume of non-implausible input space, with w indicating the history matching iteration number or wave index. For each output index o , the implausibility measure $I_o(\mathbf{x})$ is a function acting on each element of the set $\mathbf{x} = \{\vec{x}^{(i)}\}$. We use the definition,

$$I_o(\mathbf{x}) = \frac{\left| f_o(\mathbf{x}, \vec{\beta}_*(\mathbf{x})) - z_o^{\text{exp}} \right|}{\sqrt{\text{Var} \left[f_o(\mathbf{x}, \vec{\beta}_*(\mathbf{x})) \right] + \text{Var} [\varepsilon_o(\mathbf{x})]}}.$$

Here, $\text{Var} \left[f_o(\mathbf{x}, \vec{\beta}_*(\mathbf{x})) \right]$ is the variance of $f_o(\mathbf{x}, \vec{\beta}_*(\mathbf{x}))$ over the set of elements $\{\vec{x}^{(i)}\}$, and $\text{Var}[\varepsilon_o(\mathbf{x})]$ corresponds to estimates of the mean squared error due to Monte Carlo stochastic noise, Euclidean time extrapolation, and infinite volume extrapolation.”

How is the likelihood that connects the prior and posterior in Fig. S3 defined? Is it a diagonal normal distribution or something else?

We thank the referee for this question. The previous version of the manuscript had some confusing language regarding prior and posterior distributions, and the figure in question, the old Fig. S3, has been replaced. If the referee is asking about the Markov Chain Monte Carlo process used to select the 2N LECs, then in the new version of the manuscript we write:

“We use history matching to filter the 2N LECs and 3N LECs that provide an acceptable match between *ab initio* calculations and experimental data for the binding energies of several selected nuclei. The first step in our analysis is to use a Markov Chain Monte Carlo (MCMC) process to obtain a distribution of 2N LECs whose phase shifts and mixing angles corresponds with the N3LO theoretical error bands in Fig. S10. We use the Metropolis algorithm,⁸⁶ and our detailed balance function is a Gaussian function of the phase shifts and mixing angles, as prescribed by the mean values and one-sigma deviations for the N3LO theoretical error bands in Fig. S10. The equilibrium distribution for the 2N LECs obtained from this MCMC process serves as the prior distribution for our history matching analysis. This starting distribution of the 2N LECs is shown as the filled blue bars in Fig. S12, and the correlations are shown in the left panel of Fig. S13.”

But if the referee’s question is instead asking about the history matching process and the likelihood

associated with the implausibility criterion, then in the new version of the manuscript we write:

“Any particular element $\vec{x}^{(i)}$ of the set \mathbf{x} that gives a large value for I_o is considered implausible. This means that this input is unlikely to produce an acceptable match between the calculated outputs and experimental data. For this purpose, we define the maximum implausibility measure $I_M(\mathbf{x})$ as

$$I_M(\mathbf{x}) = \max_{o \in Z_w} I_o(\mathbf{x}),$$

where Z_w is again the set of outputs that we consider in the w^{th} iteration, or wave index w . We discard the input $\vec{x}^{(i)}$ if $I_M(\vec{x}^{(i)}) > c$, where c is our cutoff threshold for non-implausibility. Following Pukelsheim’s 3–sigma rule,⁸⁷ we use $c = 3$. This corresponds to 95% of the probability distribution lying within $\pm 3\sigma$ of the mean value for a random probability distribution with variance σ .”

So the likelihood function used here is a simple step function that equals zero for the implausible data points and equals one for the non-implausible data points.

In the caption of Fig. S3, it says that the prior (solid red line) gives the theoretical error bands in Fig. S2. On p.9 it says that “[...] use the theoretical errors shown in Fig. S2 and get posterior distributions of [...] and the results are shown in Fig. S3.”. I believe I understand how you obtain the posterior, but it seems like the posterior and prior are the same. Additionally, how was the prior determined?

We thank the referee for this question. As noted above, the previous version of the manuscript had some confusing language regarding prior and posterior distributions. The analysis has been redone and the figure in question, the old Fig. S3, has been replaced. The new figure is Fig. S12, and the figure caption now states:

“The filled blue bars are prior distributions obtained from MCMC sampling of the 2N LECs according to the N3LO theoretical error bands in Fig. S10. The hollow red bars are the final posterior distributions for the 2N LECs obtained after four waves of the history matching analysis.”

We have also added further details about the determination of the theoretical error bars:

“In our calculations we determine the LECs of the 2N chiral interactions at N3LO by fitting the calculated neutron-proton scattering phase shifts and mixing angles on the lattice to the Nijmegen PWA (NPWA).⁶⁷ The fundamental sources of uncertainties on these LECs are the systematic errors due to the truncated chiral EFT expansion and the statistical errors provided by the NPWA. To comprehensively estimate both systematic and statistical uncertainties, we follow the methodology introduced in Refs.^{84,85} The approach involves two components and in the first one, we define the systematic errors of an observable $X(p)$ due to the truncated chiral EFT expansion at order $N^{\text{m}}\text{LO}$ and momentum p using the following formula ...”

Reviewer Reports on the Second Revision:

Referees' comments:

Referee #1 (Remarks to the Author):

The amount of extra work and analysis the authors have invested since the last report is really appreciated. The additional results address and resolve the issues raised in the previous reports. In conclusion the current manuscript has significantly improved, several issues have been resolved, and clarifications added. It clearly demonstrates the novelty of the wave-function matching method and it should be interesting to the broad readership of Nature.

Referee #2 (Remarks to the Author):

Dear Editor,

I extend my sincerest gratitude to the distinguished authors for their diligent efforts in addressing my inquiries regarding their manuscript. I am pleased to acknowledge the significant improvements made to the "Method" section, which now exhibits enhanced clarity and logical coherence. Particularly noteworthy is the inclusion of Section S9, which presents a valuable benchmark for evaluating wave-function matching, thereby enriching the discourse on the transformation of the Hamiltonian. This is important, not only within the realm of nuclear physics but also for numerous fields in quantum physics.

Regrettably, while commendable progress has been achieved, the overall presentation of the five-page manuscript still suffers from ambiguity and lack of structural organization. This concern is exemplified by the authors' treatment of the Hamiltonian concept. They first mention H and H^A as, respectively, the high-fidelity Hamiltonian and a simple Hamiltonian in a way that suggests they correspond to many-body Hamiltonians. However in the paragraph below Fig. 1, they explain that « For [their] realistic Hamiltonian H , [they] use χ EFT two-nucleon interactions at N³LO... » Since later they indicate that the many-body problem is solved including three-body forces, H and H^A should be understood as only two-body Hamiltonians. To allow the text to be understood by the broad readership of Nature, both concepts should be clearly disentangled, e.g., using different notations for two- and many-body Hamiltonians.

Furthermore, the pivotal aspect of wave-function matching, central to the study's methodology, remains inadequately presented within the main body of the manuscript, relegating crucial insights to Section S4. Such omission hinders comprehension and necessitates undue reliance on conjectural explanations. Notably, extraneous details persist in the main narrative, further exacerbating the issue by obscuring the delineation between essential methodological intricacies and peripheral information. For example, the third paragraph of page 3 should probably be moved to the « Method » section.

As already stated in my previous report, « scientific publishing serves the purpose of disseminating novel findings to the community. Due to the wide readership of Nature, clarity and precision in communication are paramount. » It seems that the authors hold divergent perspectives. If Nature does not care about the form of the manuscripts it publishes, then this paper might be accepted for publication. Otherwise, I recommend a rejection.

Referee #3 (Remarks to the Author):

The manuscript has undergone significant improvements and shows very promising progress and potential for publication in Nature.

The revised methods section better articulates the calibration process. This new clarity, however, introduces a few additional concerns regarding the calibration and uncertainty analysis, which I believe merit further clarification by the authors. Addressing this will not only strengthen the manuscript's scientific rigor but also enhance its accessibility and comprehension for the journal's readership.

My remaining comments and concerns:

1. Concerns regarding MCMC sampling and history matching:

a) The irregularity of the distributions displayed in the left panel of Fig. S13 casts some doubt on the convergence of the MCMC sampling. Can the authors provide an explanation or address potential worries here?

b) Reading the new text that more clearly describes the calibration process raises some new concerns about the history matching method used for determining 2N and 3N LEC values. The manuscript inconsistently refers to the results of this method as posterior values in some sections and in Figs. S12 (red hollow bars) and Fig. S13 (right panel), and as non-implausible volumes in others part of the text. Given that history matching typically does not yield a posterior probability density, clarification is needed on whether these are indeed posteriors, or if this discrepancy is due to language carryover from previous manuscript versions.

c) A detailed description of the distribution of also the 3N LEC values in parameter space following history matching would be

informative. Specifically, it would be interesting to know if there is any significant tension between the 3N LEC values in the final wave and the imposed 30% restriction on the level of expectation values.

2. Some clarifications in terminology and methodology:

a) The term 'history matching uncertainties' mentioned in Tables S7 and S9 requires a clear definition in the text, considering its importance to the overall discussion.

b) The manuscript should also clarify the concept of 'best fit' values for 2N and 3N LECs. It is currently somewhat ambiguous whether these values, presumably those that best reproduce experimental energies, are the ones applied in various calculations, such as in Table S9 for charge radii.

3. Suggestion for improvements in presentation:

a) Enhancing the readability of Table S7 by indicating (possibly using bold or italic formatting) which nuclei were utilized as data in the respective waves would aid the reader.

Author Rebuttals to Second Revision:

Reply to Reviewer 1

The amount of extra work and analysis the authors have invested since the last report is really appreciated. The additional results address and resolve the issues raised in the previous reports. In conclusion the current manuscript has significantly improved, several issues have been resolved, and clarifications added. It clearly demonstrates the novelty of the wave-function matching method and it should be interesting to the broad readership of Nature.

We are grateful to the referee for the positive comments and for their help during the review process.

Reply to Reviewer 2

Dear Editor,

I extend my sincerest gratitude to the distinguished authors for their diligent efforts in addressing my inquiries regarding their manuscript. I am pleased to acknowledge the significant improvements made to the "Method" section, which now exhibits enhanced clarity and logical coherence. Particularly noteworthy is the inclusion of Section S9, which presents a valuable benchmark for evaluating wave-function matching, thereby enriching the discourse on the transformation of the Hamiltonian. This is important, not only within the realm of nuclear physics but also for numerous fields in quantum physics.

Regrettably, while commendable progress has been achieved, the overall presentation of the five-page manuscript still suffers from ambiguity and lack of structural organization. This concern is exemplified by the authors' treatment of the Hamiltonian concept. They first mention H and H^S as, respectively, the high-fidelity Hamiltonian and a simple Hamiltonian in a way that suggests they correspond to many-body Hamiltonians. However in the paragraph below Fig. 1, they explain that « For [their] realistic Hamiltonian H , [they] use EFT two-nucleon interactions at N³LO... » Since later they indicate that the many-body problem is solved including three-body forces, H and H^S should be understood as only two-body Hamiltonians. To allow the text to be understood by the broad readership of Nature, both concepts should be clearly disentangled, e.g., using different notations for two- and many-body Hamiltonians.

Furthermore, the pivotal aspect of wave-function matching, central to the study's methodology, remains inadequately presented within the main body of the manuscript, relegating crucial insights to Section S4. Such omission hinders comprehension and necessitates undue reliance on conjectural explanations. Notably, extraneous details persist in the main narrative, further exacerbating the issue by obscuring the delineation between essential methodological intricacies and peripheral information. For example, the third paragraph of page 3 should probably be moved to the Method section.

As already stated in my previous report, scientific publishing serves the purpose of disseminating novel findings to the community. Due to the wide readership of Nature, clarity and precision in communication are paramount. It seems that the authors hold divergent perspectives. If Nature does not care about the form of the manuscripts it publishes, then this paper might be accepted for publication. Otherwise, I recommend a rejection.

Following the referee’s suggestions, we have made significant improvements to the main text. We explain that wave function matching is performed at the two-body level and that the interactions are two-body interactions. We also state the key concept of wave function matching... that $\psi'_0(r)$ and $\psi_0^S(r)$ are numerically close to each other and this accelerates the convergence of perturbation theory in powers of $H' - H^S$. We now write:

“While keeping the observable physics unchanged, wave function matching creates a new high-fidelity Hamiltonian H' such that the two-body wave functions up to some finite range match that of a simple Hamiltonian H^S which is easily computed. This allows for a rapidly converging expansion in powers of the difference $H' - H^S$. Wave function matching can be used with any computational scheme. In the following analysis, we focus on the case of quantum Monte Carlo simulations, where the method presents a promising and practical strategy for evading the sign problem in realistic calculations of nuclear quantum many-body systems. While H^S and H' act on many-body systems, the wave function matching process is done at the two-body level only. For the sake of clarity, we will therefore view H^S and H' as containing only two-body interactions. Later we will also consider the inclusion of three-body interactions. However, that analysis is separate from wave function matching.

A unitary transformation U is a linear transformation that maps normalized orthogonal states to other normalized orthogonal states. Starting from a high-fidelity Hamiltonian H with only two-body interactions, wave function matching defines a new Hamiltonian $H' = U^\dagger H U$, where U^\dagger is the Hermitian conjugate of U . The unitary transformation is performed at the two-body level. In each two-body angular momentum channel, the unitary transformation U is active only when the separation distance between two particles is less than some chosen distance R . For the calculations presented here, the value $R = 3.72$ fm is used. The dependence on R is extensively discussed in Methods.

Let us write $\psi_0(r)$, $\psi'_0(r)$, and $\psi_0^S(r)$ for the two-body ground state wave functions of H , H' , and the simple Hamiltonian H^S , respectively. The transformation U is defined such that $\psi'_0(r)$ is proportional to $\psi_0^S(r)$ for $r < R$. The simple Hamiltonian is chosen so that the constant of proportionality is close to 1. For $r > R$, however, U is not active and so $\psi'_0(r)$ remains equal $\psi_0(r)$. The key point to notice here is that $\psi'_0(r)$ and $\psi_0^S(r)$ are numerically close to each other for all values of r . This can be seen visually in the left panel of Fig. 1 and is the reason why perturbation theory in powers of $H' - H^S$ converges quickly when starting from low-energy states of H^S .”

The third paragraph of the third page contains key information needed for the reader to understand our intuition and treatment of the three-nucleon interactions. While this is not directly related to wave function matching, it is an important part of the nuclear structure calculations presented in this work. We have therefore removed some extraneous details and streamlined the paragraph to half its original length. We now write:

“The results in Ref.^{28,29} showed that the range and locality of the nuclear interactions have a strong influence on nuclear binding and that the $\alpha\alpha$ interaction is highly sensitive to the range and locality of the nucleonic interactions as well as omitted higher-order interactions. These same arguments apply to other interactions involving α particles and nucleons. In Methods, we use the formalism of cluster effective field theory^{30–33} for α particles and nucleons to provide a simple counting argument for the number of parameters that require tuning to reduce unwanted errors. Our strategy is to tune the short-distance features of the three-nucleon interactions to achieve this error cancellation. We should emphasize that

our calculations are full A -body calculations, and cluster effective field theory is only used to diagnose sensitivities to short-distance physics.”

Reply to Reviewer 3

The manuscript has undergone significant improvements and shows very promising progress and potential for publication in Nature.

The revised methods section better articulates the calibration process. This new clarity, however, introduces a few additional concerns regarding the calibration and uncertainty analysis, which I believe merit further clarification by the authors. Addressing this will not only strengthen the manuscript’s scientific rigor but also enhance its accessibility and comprehension for the journal’s readership.

My remaining comments and concerns:

1. Concerns regarding MCMC sampling and history matching:

a) The irregularity of the distributions displayed in the left panel of Fig. S13 casts some doubt on the convergence of the MCMC sampling. Can the authors provide an explanation or address potential worries here?

We thank the referee for pointing out the issue of irregular shapes. There were no problems with the data at all. The irregular shapes in the left panel of Fig. S12 were caused by non-optimal choices made when drawing the correlation plots. We have redrawn the data with some additional data points as well as new contour level values that are less sensitive to small stochastic fluctuations. The shapes are no longer irregular.

b) Reading the new text that more clearly describes the calibration process raises some new concerns about the history matching method used for determining $2N$ and $3N$ LEC values. The manuscript inconsistently refers to the results of this method as posterior values in some sections and in Figs. S12 (red hollow bars) and Fig. S13 (right panel), and as non-implausible volumes in others part of the text. Given that history matching typically does not yield a posterior probability density, clarification is needed on whether these are indeed posteriors, or if this discrepancy is due to language carryover from previous manuscript versions.

We thank the referee for pointing out the wrong use of terminology and inconsistency. In the text describing Fig. S12, we write:

“The $2N$ LECs corresponding to the non-implausible volume G_4 are displayed as hollow red bars in Fig. S12. The filled blue bars in Fig. S12 are the prior distributions obtained from MCMC sampling of the $2N$ LECs according to the N^3 LO theoretical error bands in Fig. S10, and the dashed lines give the best fit values for the $2N$ LECs. The corresponding correlations are shown in Fig. S13.”

The caption for Fig. S12 has been changed to:

“Prior distributions and final non-improbable volumes for the 2N LECs. The filled blue bars are prior distributions obtained from MCMC sampling of the 2N LECs according to the N3LO theoretical error bands in Fig. S10. The hollow red bars are the non-improbable volumes for the 2N LECs obtained after four waves of the history matching analysis.”

The caption for Fig. S13 now reads:

“Prior and final non-improbable volume correlations for the 2N LECs. The left panel shows correlations in the prior distributions obtained from MCMC sampling of the 2N LECs according to the N3LO theoretical error bands in Fig. S10. The right panel shows correlations in the non-improbable volumes for the 2N LECs obtained after four waves of the history matching analysis. ”

c) A detailed description of the distribution of also the 3N LEC values in parameter space following history matching would be informative. Specifically, it would be interesting to know if there is any significant tension between the 3N LEC values in the final wave and the imposed 30% restriction on the level of expectation values.

We thank the referee for this useful suggestion. We have added the requested new figure, Fig. S14. In the revised text we write:

“The values are restricted to ensure that the expectation values of individual 3N interactions do not exceed about 30% of the total contribution from the expectation value of the 2N interactions. This constraint is enforced by setting maximum and minimum values for the individual 3N LECs and prevents unphysically large 3N energy cancellations among the different 3N interactions. In our numerical experiments, we observed a significant adverse impact on the results for the excited states in the absence of such a constraint.”

Later we write:

“In Fig. S14 we plot the distributions for the 3N coefficients corresponding to the non-improbable volume after WAVE 4. The horizontal axis is in lattice units. The dashed lines are 3N coefficient values that produce the best fit for the fitted binding energies indicated in the right panel of Fig. 2, when using 2N LECs set to the best fit values for the scattering phase shifts and mixing angles. In some cases, we can see the importance of the maximum and minimum boundary constraints on the 3N interaction coefficients. See the discussion about constraints on the 3N interaction coefficients directly after Eq. (S54).”

2. Some clarifications in terminology and methodology:

a) The term ‘history matching uncertainties’ mentioned in Tables S7 and S9 requires a clear definition in the text, considering its importance to the overall discussion.

In the text describing Table S7, we now write:

“The central values and uncertainty estimates correspond to the mean values and standard deviations

of the binding energies associated with points in the non-implausible volume.”

In the text describing Table S9, we now write:

“The central values and uncertainty estimates correspond to the mean values and standard deviations of the radii associated with points in the non-implausible volume.”

In the caption of Table S9, we now write:

“The central values and error bars are the mean values and standard deviations of the radii associated with points in the non-implausible volume.”

In the text describing Table S10, we now write:

“For the energies and several charge radii where the history matching analysis is performed, the corresponding uncertainties from the history matching analysis are shown as a second set of quoted errors. These uncertainties are the standard deviations of the binding energies and radii associated with points in the non-implausible volume.”

b) The manuscript should also clarify the concept of 'best fit' values for 2N and 3N LECs. It is currently somewhat ambiguous whether these values, presumably those that best reproduce experimental energies, are the ones applied in various calculations, such as in Table S9 for charge radii.

The referee mentions Table S9, but the best fit 2N and 3N LECs are not used in Table S9. The new caption for Table S9 clarifies this:

“The central values and error bars are the mean values and standard deviations of the radii associated with points in the non-implausible volume.”

The best fit values for the 2N and 3N LECs are used in Table S10. In the text describing Table S10, we write:

“The central values presented are obtained using the best fit values for the 2N and 3N coefficients. By this we mean the 2N coefficients that give the best fit for the scattering phase shifts and mixing angles, together with the 3N coefficients that give the best fit for the fitted nuclear binding energies indicated in the right panel of Fig. 2.”

In the caption of Fig. S10, we also write:

“For all the calculated energies and radii, the quoted central values are computed using 2N LECs set to the best fit values for the scattering phase shifts and mixing angles, together with the 3N coefficients that produce the best fit for the fitted binding energies indicated in the right panel of Fig. 2.”

3. Suggestion for improvements in presentation:

a) Enhancing the readability of Table S7 by indicating (possibly using bold or italic formatting) which nuclei were utilized as data in the respective waves would aid the reader.

This is a very good suggestion. In the new Table S7, the nuclear binding energies used in the fit for each wave are indicated with bold font.

Reviewer Reports on the Third Revision:

Referees' comments:

Referee #3 (Remarks to the Author):

I sincerely thank the authors for their significant efforts to improve the manuscript and providing detailed replies to all my comments and questions. Overall, I consider this novel work to be very creative and the results to be of outstanding scientific importance to the broad readership of Nature.